# The dynamic interplay between ATP/ADP levels and autophagy sustain neuronal migration in vivo

Cedric Bressan[1,2], Alessandra Pecora[1,2], Dave Gagnon[1,2], Marina Snapyan[1,2], Simon Labrecque[1,2], Paul De Koninck[1,2], Martin Parent[1,2], Armen Saghatelyan[1,2]*

[1]CERVO Brain Research Center, Quebec City, Canada; [2]Université Laval, Quebec City, Canada

**Abstract** Cell migration is a dynamic process that entails extensive protein synthesis and recycling, structural remodeling, and considerable bioenergetic demand. Autophagy is one of the pathways that maintain cellular homeostasis. Time-lapse imaging of autophagosomes and ATP/ADP levels in migrating cells in the rostral migratory stream of mouse revealed that decreases in ATP levels force cells into the stationary phase and induce autophagy. Pharmacological or genetic impairments of autophagy in neuroblasts using either bafilomycin, inducible conditional mice, or CRISPR/Cas9 gene editing decreased cell migration due to the longer duration of the stationary phase. Autophagy is modulated in response to migration-promoting and inhibiting molecular cues and is required for the recycling of focal adhesions. Our results show that autophagy and energy consumption act in concert in migrating cells to dynamically regulate the pace and periodicity of the migratory and stationary phases to sustain neuronal migration.

## Introduction

Cell migration is a crucial mechanism for normal embryonic development, and a cell migration deficit leads to the loss of embryos or functional and cognitive impairments (*Ayala et al., 2007*). Although neuronal migration largely ceases in the postnatal period, it is preserved in several areas associated with postnatal neurogenesis such as the cerebellum, hippocampus, and subventricular zone (SVZ), which is the area associated with the production of neuronal precursors destined for the olfactory bulb (OB). Neuronal precursors born in the SVZ migrate tangentially along the rostral migratory stream (RMS) and, once in the OB, turn to migrate radially and individually out of the RMS into the bulbar layers (*Gengatharan et al., 2016*; *Kaneko et al., 2017*). In the postnatal RMS, neuroblasts travel in chains ensheathed by astrocytic processes along blood vessels (*Snapyan et al., 2009*; *Kaneko et al., 2010*; *Whitman et al., 2009*). Cell migration is a very dynamic process composed of migratory phases intercalated with stationary periods and is accompanied by structural remodeling, organelle dynamics, and protein trafficking and turnover (*Tanaka et al., 2017*; *Webb et al., 2002*). Cell migration also entails considerable bioenergetic demand, and it is still unclear how migrating cells regulate cellular homeostasis by clearing damaged organelles and aggregated/misfolded proteins, and how the metabolic requirements of neuroblasts are dynamically regulated during the different phases of cell migration.

Autophagy is a key evolutionarily conserved intracellular process that controls protein and organelle degradation and recycling. During this process, intracellular materials are sequestered inside double-membrane vesicles (autophagosomes) that then fuse with lysosomes to form autolysosomes in which aggregated/misfolded proteins and damaged organelles are degraded (*Yang and Klionsky, 2010*). Autophagy plays an important role in neurodevelopment (*Mizushima and Levine, 2010*) and alterations in autophagy have been observed in various

*For correspondence:
armen.saghatelyan@fmed.ulaval.ca

**Competing interests:** The authors declare that no competing interests exist.

human diseases (*Yang and Klionsky, 2010*). Autophagy is under control of a large protein complex and may be initiated by the Ulk1 and Ulk2 proteins, the downstream effectors of AMP-activated kinase (AMPK) and mTOR (*Egan et al., 2011*; *Petherick et al., 2015*). The initiation of autophagy is accompanied by the formation of Atg5–Atg12 complexes and by Atg7-dependent conversion of microtubule-associated protein light chain 3 (LC3) from the inactive cytoplasmatic form into the membrane-bound lipidated form (LC3-II) that is found at the surface of autophagosomes (*Simonsen and Tooze, 2009*). Recent studies have shown that autophagy may be involved in neuronal migration. For example, a mutation in the Vps15 gene dysregulates endosomal-lysosomal trafficking and autophagy that, in turn, perturbs neuronal migration (*Gstrein et al., 2018*). Similarly, ablation of Vps18 leads to neurodegeneration by blocking multiple vesicle transport pathways, including autophagy, and results in abnormal neuronal migration through the upregulation of β1 integrin (*Peng et al., 2012*). In the OB, knock-down of let-7 microRNA (miRNA) decreases the radial migration of neuroblasts by interfering with autophagy (*Petri et al., 2017*). Overexpression of beclin-1 or TFEB in neuroblasts lacking let-7 re-activates autophagy and restores radial migration (*Petri et al., 2017*). On the other hand, the ectopic expression of autism-related nonsense Foxp1 mutant-induced autophagy delays radial migration in the developing cortex (*Li et al., 2019*). While all these studies have shown that there is a link between neuronal migration and autophagy, genetic manipulations of Vps15, Vps18, let-7, and Foxp1 affect not only autophagy but also several other molecular pathways that may potentially impact cellular migration (*Gstrein et al., 2018*; *Peng et al., 2012*; *Petri et al., 2017*; *Li et al., 2019*). As such, the exact roles played by key autophagy genes in the migration of neuronal cells need to be determined. It is also unclear how autophagy is dynamically regulated in neuroblasts to allow them to cope with the pace and periodicity of neuronal migration, whether the level and activity of autophagy is modulated by migration-promoting and migration-inhibiting molecular cues, and how autophagy is induced.

Autophagy may be triggered by changes in the cellular bioenergetic state. The regulation of ATP/ADP levels is an important process for cells with high-energy demands such as mature neurons (*Hyder et al., 2013*; *Hill and Colón-Ramos, 2020*). The modulation of neuronal activity is accompanied by local changes in energy levels and the induction of metabolic compartments near synapses to support synaptic function (*Jang et al., 2016*). In both physiological and pathological conditions, neuronal functions may be linked to the activation of the autophagy pathway at the synaptic level where the biogenesis of autophagosomes may take place (*Hill and Colón-Ramos, 2020*). The link between energy demand and autophagy in the context of neuronal migration, another high-energy demand process, remains largely unexplored. Furthermore, it remains unclear how these two major housekeeping cellular processes dynamically interact in migrating cells to sustain long-range neuronal migration.

Here, we show that autophagy is dynamically modulated in migrating cells in the RMS and that autophagy levels change in response to molecular cues that promote or inhibit migration. Changes in autophagy levels are linked to changes in the recycling of the focal adhesion protein paxillin. We used time-lapse imaging of autophagosomes and measurements of ATP/ADP levels in migrating cells to show that, under basal conditions, autophagy levels are correlated with the bioenergetic requirements of cells and that decreases in ATP/ADP levels during the migratory phase lead to the entry of cells into the stationary phase, the activation of AMPK, and the induction of autophagy. The genetic disruption of autophagy-related proteins and downstream targets of AMPK such as Ulk1, Ulk2, Atg5, and Atg12 hampers neuronal migration by prolonging the length of the stationary phase, leading to the accumulation of cells in the RMS. Our results show that autophagy, by sustaining neuronal migration, regulates the faithful arrival of neuronal precursors into both the developing and adult OB. Our results also mechanistically link autophagy and energy consumption in migrating cells to the dynamic regulation of the pace and periodicity of the migratory and stationary phases.

# Results

## Autophagy is dynamically regulated in migrating neuroblasts

To investigate the involvement of autophagy in cell migration in vivo, we first performed immunostaining for two key autophagic proteins, microtubule-associated protein 1 light chain 3 (LC3, isoforms A and B) and Atg5, on sagittal sections of the adult mouse forebrain. Neuroblasts were labeled using GFP-expressing retroviruses injected into the SVZ (*Figure 1A*). LC3 and Atg5 were highly expressed in GFP+ migrating neuroblasts in the RMS (*Figure 1B–C*). Between 55 and 60% of the neuroblasts were immunolabeled for Atg5 or LC3 (*Figure 1B–D*). In contrast, less than 10% of astrocytes in the RMS of GFAP-GFP mice were immunolabeled for autophagy-related proteins (*Figure 1D*). To confirm the presence of autophagosome vesicles, which are characterized by a double membrane (*Klionsky et al., 2012*), we examined RMS samples by electron microscopy. We took advantage of tamoxifen-inducible *Slc1a3*(*Glast*)<sup>CreErt2</sup>::GFP mice in which GFP is expressed in astrocytes and stem cells and their progeny following tamoxifen injection. GFP+-immunolabeled neuroblasts, which were identified based on their typical bipolar

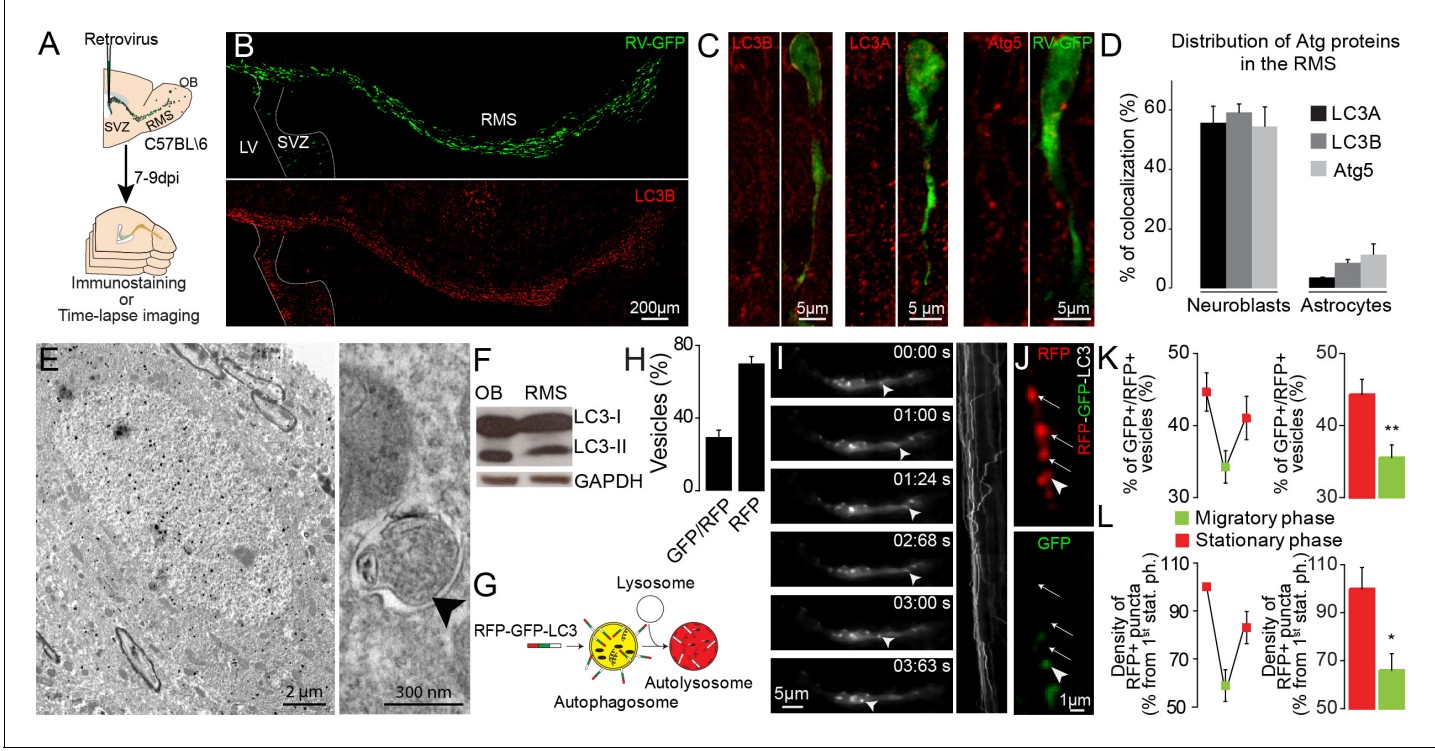

**Figure 1.** Migrating cells express autophagy-related proteins and display active autophagic flux. (**A**) Experimental design for the quantification of autophagy in migrating neuroblasts (**B–D**) Confocal images and quantification showing Atg5, LC3A, and LC3B expression by neuroblasts and astrocytes. The percentage of co-localization was calculated as the number of GFP+ cells (neuroblasts or astrocytes) with labeling for one of these proteins over the total number of GFP+ cells. (**E**) Electron microscopy (EM) analysis of autophagy in neuroblasts in the RMS of *Slc1a3*(*Glast*)<sup>CreErt2</sup>::GFP mice. The neuroblasts were labeled with anti-GFP immunogold particles. An autophagosome is indicated with an arrowhead. (**F**) Western blot analysis of LC3 in OB and RMS samples. (**G**) Scheme of the RFP-GFP-LC3 fusion protein used to study autophagic flux. The RFP-GFP-LC3 fusion protein makes it possible to label autophagosomes for RFP and GFP. Autolysosomes were only labeled for RFP because GFP was quenched due to the acidic pH in the lysosomes. (**H**) Percentage of GFP+/RFP+ vesicles with respect to RFP+ vesicles in neuroblasts in fixed brain sections obtained after stereotaxic injections of retrovirus-encoding RFP-GFP-LC3 (n = 10 cells from three animals) (**I**) Wide-field time-lapse imaging and kymograph of RFP-LC3 vesicles in the leading process of migrating neuroblasts. (**J**) Example of two-photon imaging of RFP-GFP-LC3 vesicles in the leading process of migrating neuroblasts in acute brain sections. RFP+ autophagosomes/autolysosomes are indicated with arrows, while GFP+/RFP+ autophagosome is indicated with an arrowhead. (**K:**) Quantification of GFP+/RFP+ vesicles among RFP+ vesicles during the different phases of cell migration (n = 18 cells from three animals). Data are expressed as means ± SEM, **p<0.005 with the Student t-test. (**L**) Quantification of changes in the RFP+ punctae of individual neuroblasts during the different phases of cell migration. Data are plotted as the density of RFP+ autophagosomes/autolysosomes in neuroblasts during the different migration phases (n = 11 cells from six animals). The results were normalized to the density of the first stationary phase to be able to make inter-cell and inter-animal comparisons. Data are expressed as means ± SEM, *p<0.05 with the Student t-test. See also *Video 1*.

morphology with leading and trailing processes, exhibited numerous double-membrane autophagosomes (*Figure 1E*). LC3 is present in a non-lipidated form (LC3-I) in the cytoplasm and a conjugated phosphatidylethanolamine lipidated form (LC3-II) in the autophagosomal membrane (*Kabeya et al., 2004*). The presence of LC3-II is a sign of autophagosome formation (*Klionsky et al., 2012*). Western blots of micro-dissected OB and RMS, two regions that contain migrating neuroblasts, showed the presence of autophagosome-associated LC3-II (*Figure 1F*). The presence of autophagosomes in neuroblasts could be due to active autophagic flux or interrupted autophagy such as the absence of fusion of autophagosomes with lysosomes, leading to vesicle accumulation. To distinguish between these two possibilities, we infected neuroblasts with a retrovirus encoding the RFP-GFP-LC3 fusion protein. The expression of RFP-GFP-LC3 makes it possible to distinguish autophagosomes (RFP+/GFP+ vesicles) from autolysosomes (RFP+ vesicles) due to GFP quenching in the acidic lysosomal environment. We observed a higher proportion of RFP+ than of GFP+/RFP+ vesicles and a highly dynamic bidirectional movement of RFP-LC3 punctae (*Figure 1G–I*; *Video 1*). Interestingly, 1 hr time-lapse imaging of migrating neuroblasts infected with the RFP-GFP-LC3 retrovirus revealed a dynamic regulation and non-homogeneous distribution of autophagosomes/autolysosomes during the different cell migration phases (*Figure 1K–L*). The migratory and stationary phases of neuroblasts were associated, respectively, with lower and higher numbers of GFP+/RFP+ autophagosomes with respect to RFP+ autophagosomes/autolysosomes (35.5 ± 3.1% and 44.4 ± 3.1% for the migratory and stationary phases, respectively; p<0.005) (*Figure 1K*). Moreover, the density of RFP+ punctae was lower during the migratory phases, whereas the entry of cells into the stationary phase led to a higher density of autophagosomes/autolysosomes (100 ± 8.8% and 66 ± 6.9% for the stationary and migratory phases, respectively; p<0.05) (*Figure 1L*). Altogether, these results suggest for an active autophagic flux in the migratory neuroblasts which is dynamically modulated during the different phases of cell migration.

## The suppression of autophagy hampers cell migration and leads to the accumulation of neuroblasts in the RMS

To investigate the role of autophagy in cell migration we first assessed the migration of GFP+ neuroblasts in the RMS following the application of bafilomycin (4 µM), which is known to inhibit the fusion of autophagosomes with lysosomes (*Yamamoto et al., 1998*). We stereotactically injected a GFP-encoding retrovirus in the SVZ and analyzed cell migration in the RMS 7–10 days later (*Figure 2A*). Time-lapse imaging of migrating neuroblasts in acute brain sections revealed that bafilomycin reduces the distance of migration of GFP+ neuroblasts (46.1 ± 2.9 µm for the control vs. 27.6 ± 3.3 µm following the application of bafilomycin, p<0.001, n = 55 and 41 cells from three mice per condition, respectively; *Figure 2B–C*). The decrease in migration distance was caused by changes in the percentage of the migratory phases (53.2 ± 1.6% and 33.8 ± 1.8% in the control and bafilomycin-treated cells, respectively, p<0.001, *Figure 2E*). There was no change in the speed of migration, which was estimated exclusively during the migratory phases (114.9 ± 5.0 µm/h and 97.6 ± 5.6 µm/h in control and bafilomycin-treated cells, respectively, *Figure 2D*). In these and all other migration experiments, we did not take into consideration cells that remained stationary during the entire imaging period, which likely underestimate the autophagy-dependent effects on cell migration.

We next employed a genetic strategy to ablate *Atg5*, an essential autophagy gene, using the transgenic *Slc1a3*(*Glast*)<sup>CreErt2</sup>::*Atg5*<sup>fl/fl</sup>::GFP mouse model in which a tamoxifen injection causes the inducible conditional knock-out of *Atg5* (hereafter, Atg5 cKO) in stem cells and their progeny (*Figure 2F*). We first verified the efficiency of autophagy impairment by performing an EM analysis of GFP+ neuroblasts in the RMS of *Slc1a3*(*Glast*)<sup>CreErt2</sup>::*Atg5*<sup>wt/wt</sup>::GFP (Atg5 WT) and Atg5 cKO mice and showed that the area of the cells occupied by autophagosomes

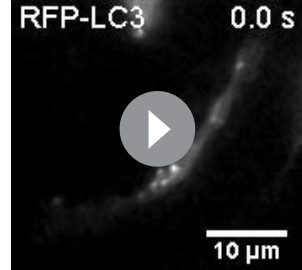

**Video 1.** Time-lapse imaging of RFP-GFP-LC3 in migrating neuroblasts. Time-lapse imaging of RFP-LC3 punctae in neuroblasts. The time is indicated in the upper left corner.
https://elifesciences.org/articles/56006#video1

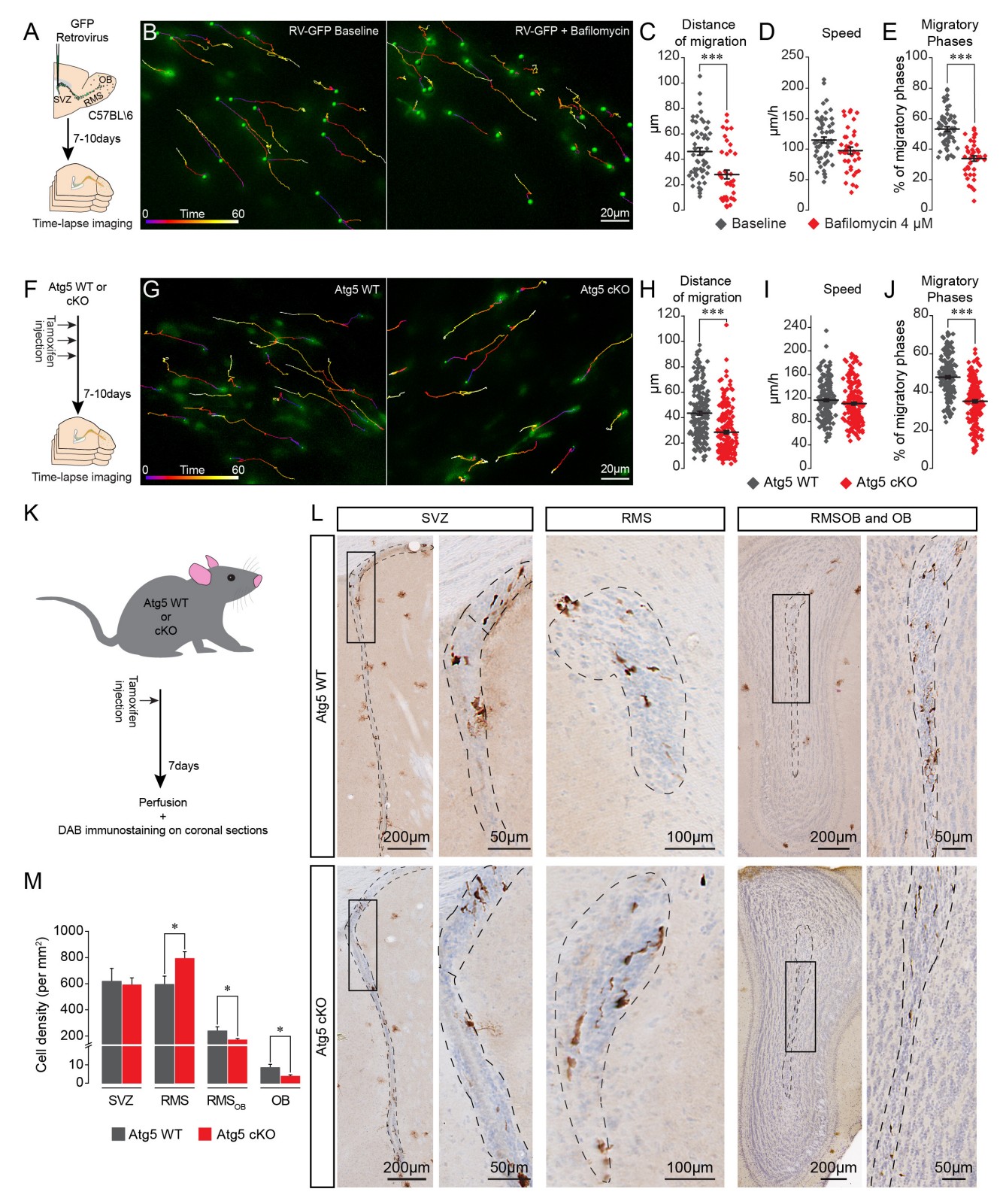

**Figure 2.** Pharmacological and genetic impairment of autophagy-related genes disrupts cell migration. (**A**) Experimental design for the pharmacological impairment of autophagy using bafilomycin. (**B**) Time-lapse imaging of GFP+ neuroblasts in acute brain sections. (**C–E**) Distance of migration, speed of migration, and percentage of migratory phases of neuroblasts under control condition and following application of bafilomycin (n = 55 and 41 cells from three mice for the control and bafilomycin mice, respectively, ***p<0.001 with the Student t-test). (**F**) Experimental design to

*Figure 2 continued on next page*

*Figure 2 continued*

conditionally delete Atg5 in adult mice. (G) Time-lapse imaging of Atg5 cKO and Atg5 WT neuroblasts in acute brain sections. (H–J) Distance of migration, speed of migration, and percentage of migratory phases of Atg5 cKO and Atg5 WT neuroblasts (n = 188 and 198 cells from 9 Atg5 WT and 9 Atg5 cKO mice, respectively. ***p<0.001 with the Student t-test). (K) Experimental design to conditionally delete Atg5 in adult mice to follow the impact of autophagy on the entire cell population. (L) Example of DAB staining of coronal sections from Atg5 cKO and Atg5 WT mice obtained from brains fixed 7 days after a single tamoxifen injection. (M) Quantification of GFP+ neuroblast density in the SVZ, RMS, RMS$_{OB}$, and OB of Atg5 WT and cKO mice (n = 8 for the Atg5 cKO mice and n = 7 for the Atg5 WT mice, *p<0.05 with the Student t-test). See also *Figure 2—figure supplement 1* and *Video 2*.

The online version of this article includes the following figure supplement(s) for figure 2:

**Figure supplement 1.** Validation of autophagy impairment in Atg5 cKO mice by electron microscopy.

was 75% lower in the neuroblasts from the Atg5 cKO mice, with no change in the size of individual autophagosomes, indicating that there was a decrease in the total number of autophagosomes (*Figure 2—figure supplement 1*). These results confirmed that the impairment of autophagy in Atg5 cKO cells is efficient. Time-lapse imaging of migrating neuroblasts in acute brain sections from Atg5 WT and Atg5 cKO mice showed that Atg5-deficient neuroblasts have a defect in the distance of migration (43.4 ± 1.5 μm in control mice vs. 28.5 ± 1.3 μm, p<0.001, n = 188 and 198 cells from nine mice per condition for the Atg5 WT and cKO mice, respectively; *Figure 2G–H*, *Video 2*). This decrease was caused by changes in the percentage of the migratory phases (47.8 ± 0.8% in Atg5 WT mice vs. 35.1 ± 0.8% in Atg5 cKO mice, p<0.001), with no change in the speed of migration, which was estimated exclusively during the migratory phases (116.2 ± 2.3 μm/h in Atg5 WT mice vs. 110.4 ± 2.4 μm/h in Atg5 cKO mice; *Figure 2G–J*). To study the impact of autophagy suppression on the total neuroblast population, we performed GFP immunostaining on serial coronal sections from the OB to the SVZ of the Atg5 WT and Atg5 cKO mice 7 days after a single tamoxifen injection (*Figure 2K–L*). An Atg5 deficiency led to the accumulation of neuroblasts in the RMS close to the SVZ (594.5 ± 62.9 cell/mm$^2$ in Atg5 WT mice vs. 793.3 ± 50.9 cell/mm$^2$ in Atg5 cKO mice; p<0.05, n = 7 WT mice and n = 8 cKO mice), with an accompanying decrease in the density of neuroblasts in the rostral RMS (RMS of the OB; noted as RMS$_{OB}$) (239.2 ± 31.1 cell/mm$^2$ in Atg5 WT mice vs. 171.1 ± 9.7 cell/mm$^2$ in Atg5 cKO mice; p<0.05) and the OB (8.5 ± 1.8 cell/mm$^2$ in Atg5 WT mice vs. 3.8 ± 0.7 cell/mm$^2$ in Atg5 cKO mice, p<0.05) (*Figure 2M*). Taken together, these results show that Atg5-dependant autophagy is required to maintain a normal periodicity of the migratory and stationary phases of migration and the correct routing of neuroblasts toward the OB.

In these experiments, the expression of Atg5 was affected at the very beginning of the stem cell lineage, which may have indirectly affected the migration of the progeny derived from these stem cells. We thus stereotactically injected a mixture of retroviruses encoding Cre-mNeptune or GFP into the SVZ of *Atg5*$^{fl/fl}$ and *Atg5*$^{wt/wt}$ mice (*Figure 3A*). The retroviruses infect rapidly dividing cells such as neuroblasts but not slowly dividing stem cells. *Atg5*$^{fl/fl}$ neuroblasts expressing Cre displayed the same deficiency in migratory parameters as neuroblasts in Atg5 cKO mice (*Figure 3B–E* and *Video 3*). Importantly, neuroblasts from *Atg5*$^{wt/wt}$ mice infected either with Cre-mNeptune or GFP retroviral particles or neuroblasts from *Atg5*$^{fl/fl}$ mice infected with GFP retroviral particles displayed normal migration patterns (*Figure 3B–E*). These results suggest that the deficiency in cell migration does not arise from the progression and differentiation of the stem cell lineage.

It has been recently reported that Atg5 deletion may produce an autophagy-independent effect (*Galluzzi and Green, 2019*). To verify this possibility, we genetically impaired the expression of *Atg12*, another essential autophagy-related gene, using CRISPR/Cas9 technology. We electroporated plasmids

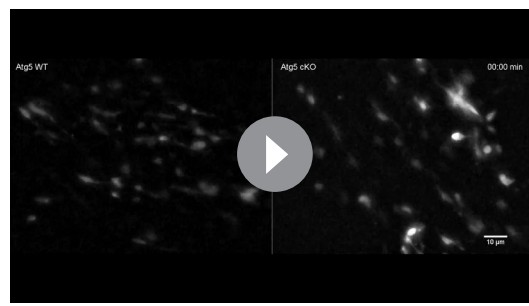

**Video 2.** Neuronal migration of Atg5 WT and Atg5 cKO neuroblasts in acute sections of the adult RMS. Time-lapse imaging of GFP+ neuroblasts in sections from Atg5 WT (left) and Atg5 cKO (right) mice. The time is indicated in the upper left corner.
https://elifesciences.org/articles/56006#video2

carrying Cas9-T2A-mCherry and *Atg12* gRNAs in the early postnatal period (*Figure 3F*). We used *LacZ* gRNAs as a control. We used HRM qRT-PCR to confirm the presence of mutated *Atg12* RNA transcripts after the infection with the gRNAs (*Figure 3—figure supplement 1*). We also confirmed the loss of the protein in vivo in *Atg12* gRNA-electroporated cells by performing immunolabeling against Atg12 in brain sections containing the SVZ and RMS (*Figure 3G*). We observed an 80% decrease in the percentage of neuroblasts expressing Atg12 and electroporated with *Atg12* gRNA as compared to *LacZ* gRNA-electroporated cells (100 ± 0% of Atg12-expressing neuroblasts in *LacZ* gRNA-electroporated cells and 20.5 ± 2.4% in *Atg12* gRNA-electroporated cells, n = 28 cells for *LacZ* gRNA and n = 43 cells for *Atg12* gRNA, three animals per group). We next performed time-lapse imaging of mCherry+ cells in the RMS 8–13 days post-electroporation and observed that *Atg12* gRNAs cause the same defects in cell migration (the distance of migration was 38.5 ± 3.3 μm for *LacZ* gRNA cells vs. 30.4 ± 2.4 μm for *Atg12* gRNA cells, p<0.05, and the percentage of migratory phases was 48.4 ± 1.7% for *LacZ* gRNA cells vs. 35.5 ± 1.5% for *Atg12* gRNA cells, p<0.001) as an Atg5 deficiency (*Figure 3H–K*). To determine whether an Atg12 deficiency also results in the accumulation of neuroblasts in the RMS close to the SVZ, we acquired images of sagittal brain sections in mice electroporated with either *LacZ* or *Atg12* gRNAs 9 days post-electroporation and quantified the density of the cells along the SVZ-OB pathway. As electroporation efficiency may vary between animals and given the fact that all the cells present in the RMS, RMS$_{OB}$, and OB were derived from cells electroporated in the SVZ, we normalized the cell density along the migratory path to the density of mCherry+ cells in the SVZ. Our analysis revealed an accumulation of *Atg12* gRNA-expressing cells in the RMS as compared to *LacZ* gRNA cells (103.4 ± 17.4% in *Atg12* gRNA mice vs. 51.5 ± 4.9% in *LacZ* gRNA mice, p<0.005), with a decreased cell density in the RMS$_{OB}$ (25.9 ± 5.4% in *Atg12* gRNA mice vs. 47.0 ± 6.5% in *LacZ* gRNA mice; p<0.05) and OB (9.2 ± 1.0% in *Atg12* gRNA mice vs. 15.4 ± 1.9% in *LacZ* gRNA mice; p<0.05, n = 8 and 7 animals for *LacZ* gRNA- and *Atg12* gRNA-electroporated mice, respectively; *Figure 3L–M*). These data are in line with those obtained with Atg5 cKO mice (*Figure 2M*) and suggest that a deficiency in either Atg5 or Atg12 leads to the accumulation of neuroblasts in the RMS proximal to the SVZ concomitant with a decrease in the number of migrating cells in the distal portions of the RMS and OB. Altogether, our results with pharmacological and genetic perturbations of autophagy in the migrating cells indicate that this self-catabolic pathway is required for cell migration.

## Autophagy is induced by bioenergetic demand during cell migration

Autophagy is associated with energetic stress (*Kroemer et al., 2010*). The energy status of cells, represented by the ATP/ADP ratio, is closely related to the ability of metastatic cancer cells to migrate in vitro (*Zanotelli et al., 2018*). We used a ratiometric biosensor of ATP/ADP (PercevalHR) (*Tantama et al., 2013*) to explore energy consumption during cell migration and to investigate the link between energy consumption and autophagy. PercevalHR- and TdTomato-encoding lentiviruses were co-injected into the SVZ of adult mice. We first validated the functionality of PercevalHR by incubating acute slices in rotenone to impair mitochondrial function and observed a concentration-dependent decrease in the ATP/ADP ratio in migrating neuroblasts (*Figure 4—figure supplement 1A–B*). We next performed time-lapse imaging of TdTomato+ and PercevalHR+ neuroblasts in the RMS in acute brain sections 5–10 days post-injection to assess changes in the ATP/ADP ratio during the different phases of cell migration (*Figure 4A* and *Video 4*). Interestingly, the ATP/ADP ratio was dynamically regulated during the different phases of cell migration (*Figure 4B–D*) similar to the dynamic changes observed for autophagosomes (*Figure 1K–L*). Shortly after the beginning of the migratory phases, the ATP/ADP ratio started to decrease. This was accompanied by the entry of the cells into the stationary phase and a progressive recovery of the ATP/ADP ratio (*Figure 4B–C*). To assess changes in the ATP/ADP ratio during the different phases of cell migration in individual cells and across different cells, the ratio was normalized to the duration of the migratory and stationary phases and was counted as changes in the charge. This analysis revealed a significant difference in ATP/ADP levels during the migratory and stationary phases of cell migration (−0.001 ± 0.001 AU in the stationary phase vs. −0.004 ± 0.001 AU in the migratory phase, p<0.05, *Figure 4D*). As the fluorescence intensity of PercevalHR is sensitive to pH (*Tantama et al., 2011*), we ensured that changes in the PercevalHR ratio did not arise from variations in the intracellular pH during the different phases of cell migration. To do so, we electroporated plasmid-encoding pHRed, a pH sensor,

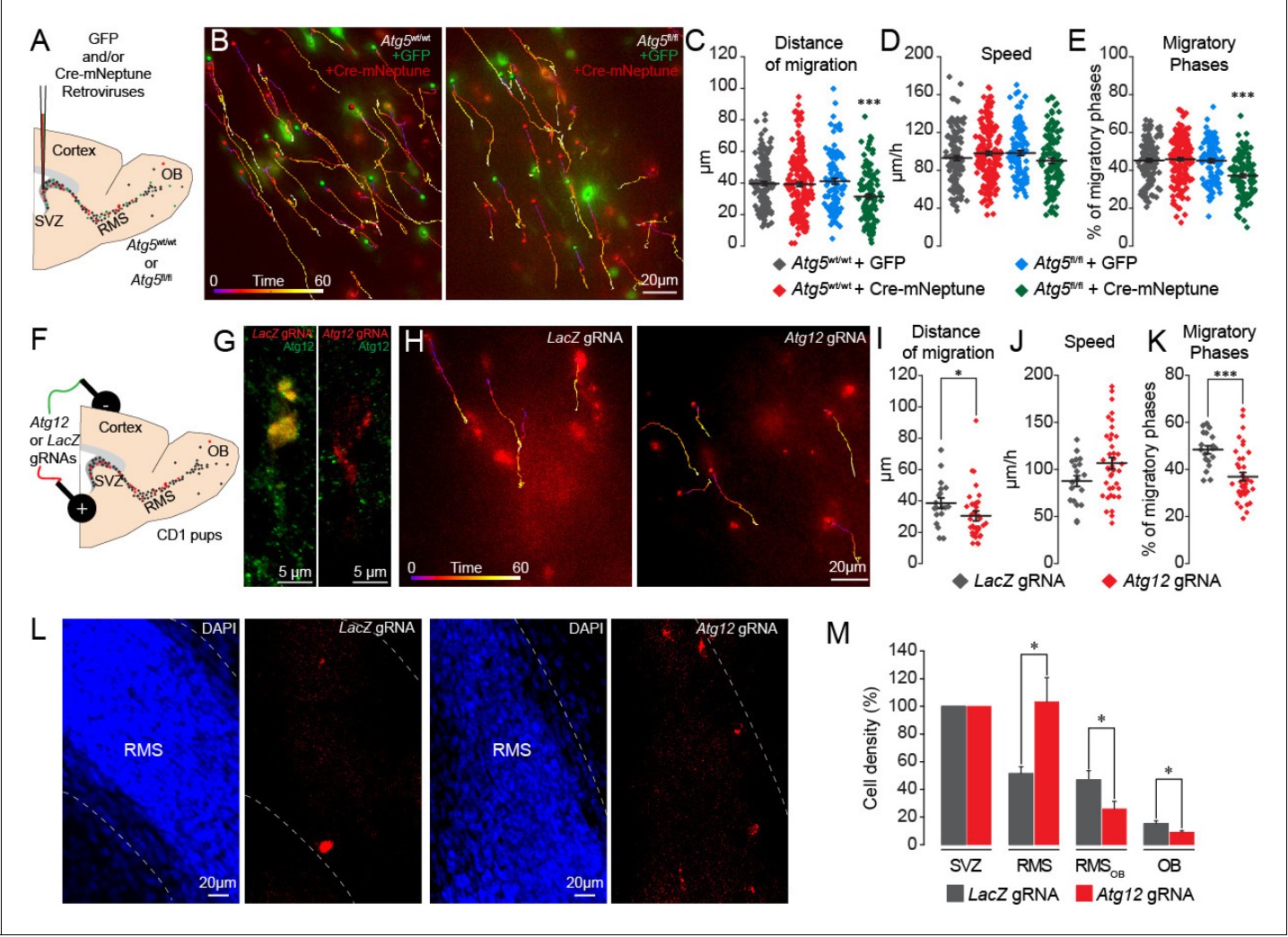

**Figure 3.** Genetic impairment of autophagy-related genes in the adult and early postnatal mice disrupts cell migration. (A) Experimental procedure for GFP- and Cre-mNeptune-encoding retroviral labeling of neuroblasts from *Atg5*wt/wt and *Atg5*fl/fl mice. (B) Example of time-lapse imaging of neuroblasts expressing GFP or Cre-mNeptune. (C–E) Distance of migration, speed of migration, and percentage of migratory phases for *Atg5*wt/wt and *Atg5*fl/fl mice injected with GFP or Cre-mNeptune retroviruses. *Atg5*wt/wt-GFP (n = 143 cells from eight mice), *Atg5*wt/wt-Cre-mNeptune (n = 208 cells from nine mice), *Atg5*fl/fl-GFP (n = 106 cells from seven mice), and *Atg5*fl/fl-Cre-mNeptune (n = 121 cells from five mice). ***p<0.001 with a one-way ANOVA followed by an LSD-Fisher post hoc test. (F) Experimental procedure for the electroporation of plasmids expressing Cas9-T2A-mCherry and gRNAs. Plasmids were injected into the lateral ventricle of P1-P2 pups followed by the application of electrical pulses. Acute sections were prepared 8–13 days post-electroporation, and the migration of electroporated cells was assessed by time-lapse imaging. (G) Example of immunostaining of Atg12 on cells expressing *LacZ* gRNA (green) and *Atg12* gRNA (red). (H) Time-lapse imaging of neuroblasts electroporated with *LacZ* gRNA or *Atg12* gRNA in acute brain sections. (I–K) Distance of migration, speed of migration, and percentage of migratory phases of cells electroporated with *LacZ* gRNA or *Atg12* gRNA (n = 19 and 40 cells from 5 and 13 animals for *LacZ* gRNA and *Atg12* gRNA, respectively, *p<0.05 and ***p<0.001 with Student t-test). Individual values and means ± SEM for all time-lapse imaging experiments are shown. (L) Example of a brain section showing neuroblasts electroporated with *LacZ* gRNA (left) or *Atg12* gRNA (right) in the RMS. (M) Quantification of Cas9-T2A-mCherry+ neuroblast density in the SVZ, RMS, RMS_OB, and OB of *LacZ* gRNA- and *Atg12* gRNA-electroporated mice. Data are expressed as a percentage of the cell density with 100% defined as the cell density in the SVZ (n = 8 mice for *LacZ* gRNA and seven mice for *Atg12* gRNA, *p<0.05). See also *Figure 3—figure supplement 1* and *Video 3*.

The online version of this article includes the following figure supplement(s) for figure 3:

**Figure supplement 1.** Validation of gRNA efficiency by high-resolution melting (HRM) PCR.

prepared acute brains sections 5–7 days post-electroporation, and monitored changes in pH during the different phases of cell migration (*Figure 4—figure supplement 1C*). Our time-lapse imaging data suggested that changes in the ATP/ADP ratio during the different phases of cell migration, as determined with the PercevalHR sensor, are not due to changes in the intracellular pH (*Figure 4—*

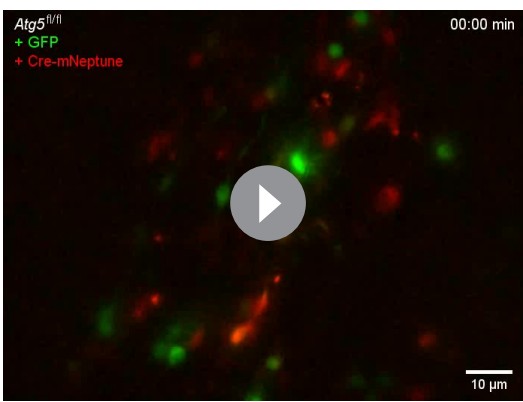

**Video 3.** Neuronal migration of Cre-mNeptune- and GFP-infected cells in *Atg5*^fl/fl^ mice. Example of time-lapse imaging of neuroblasts on acute slices from *Atg5*^fl/fl^ mice infected with retroviruses expressing GFP and Cre-mNeptune. The images were acquired every 15 s for 1 hr. The time is indicated in the upper left corner.

https://elifesciences.org/articles/56006#video3

figure supplement 1D). These results indicate that the ATP/ADP ratio is dynamically regulated during cell migration and suggest the presence of mechanisms leading to the recovery of ATP levels following the decrease during the migratory phases.

The main sensor of energy levels in cells is AMPK, which is phosphorylated when the ATP/ADP ratio decreases (*Hardie et al., 2012*). The phosphorylation of AMPK may induce autophagy (*Egan et al., 2011*) by phosphorylating Ulk1 and Ulk2 (*Klionsky et al., 2012*), which may mechanistically link the observed dynamic changes in the autophagosomes and the ATP/ADP ratio during the migratory and stationary phases. To assess the link between energy consumption and autophagy in the migrating cells we used genetic and pharmacological approaches. Since the phosphorylation of Ulk1 and Ulk2 by AMPK is crucial for the induction of autophagy, we first electroporated plasmids carrying Cas9-T2A-mCherry and *Ulk1* and *Ulk2* gRNAs (*LacZ* gRNAs were used as a control) in the early postnatal period (*Figure 4E–I*). We used HRM qRT-PCR to confirm the presence of mutated *Ulk1* and *Ulk2* RNAs transcripts after the infection with the gRNAs (*Figure 4—figure supplement 1*). We observed a reduced distance of migration for neuroblasts electroporated with *Ulk1* or *Ulk2* gRNAs (*Figure 4F–G*) because of reduced migratory phases (*Figure 4I*), with no any effect on the speed of migration (*Figure 4H*). To investigate further the interplay between energy consumption and autophagy in the migrating cells, we used a pharmacological approach to block AMPK using Compound C (cC) (*Kim et al., 2011*) in control and Atg5-deficient neuroblasts. We hypothesized that if AMPK induces the activation of autophagy, then blocking this kinase with cC in autophagy-deficient neuroblasts should not have any additional effect. Acute sections were prepared from *Slc1a3*(*Glast*)^CreErt2^::GFP mice following a tamoxifen injection, and the cells were imaged in the presence of cC (*Figure 4J–K*). cC reduced the distance of cell migration because of a decrease in the percentage of the migratory phases. The speed of migration did not change (*Figure 4K–N*). We next inhibited AMPK with cC in acute sections from Atg5 cKO mice and observed no additional changes in the distance of migration or in the duration of the migratory phases (*Figure 4K–N*). These results highlight the importance of AMPK in controlling the periodicity of the migratory and stationary phases during cell migration and indicate that changes in the energy status of cells induce autophagy.

## Autophagy activation is necessary to maintain focal adhesion recycling

We next investigated how autophagy affects cell migration. Previous studies have shown that paxillin, a focal adhesion protein, is a direct target of LC3-II and is recycled by autophagy during the migration of non-neuronal cells in vitro (*Sharifi et al., 2016*; *Kenific et al., 2016*). As the induction of autophagy and cargo recycling is cell- and context-dependent (*Klionsky et al., 2012*), we wondered whether paxillin was also recycled by autophagy in migrating neuroblasts in vivo. We electroporated plasmids carrying paxillin-GFP and RFP-LC3 in pups and analyzed their co-localization 5 days later in the RMS (*Figure 5A*). Our analysis revealed that 24.5 ± 2% to 28.4 ± 1.7% of paxillin-GFP co-localized in RFP-LC3 vesicles in the cell body and the proximal part of the leading process in WT mice, while 42.5 ± 7.9% of paxillin-GFP co-localized in the distal part of the leading process (n = 30 cells from three mice; *Figure 5B*), suggesting that paxillin is more actively recycled at the leading edge of migrating neuroblasts in the RMS. To determine whether a lack of autophagy leads to altered recycling of paxillin, we performed immunolabeling for paxillin in Atg5 WT and Atg5 cKO mice. We observed that a deficiency in autophagy leads to increased immunolabeling of paxillin in the leading process of neuroblasts, with no changes in the cell soma (n = 38 cells from three mice

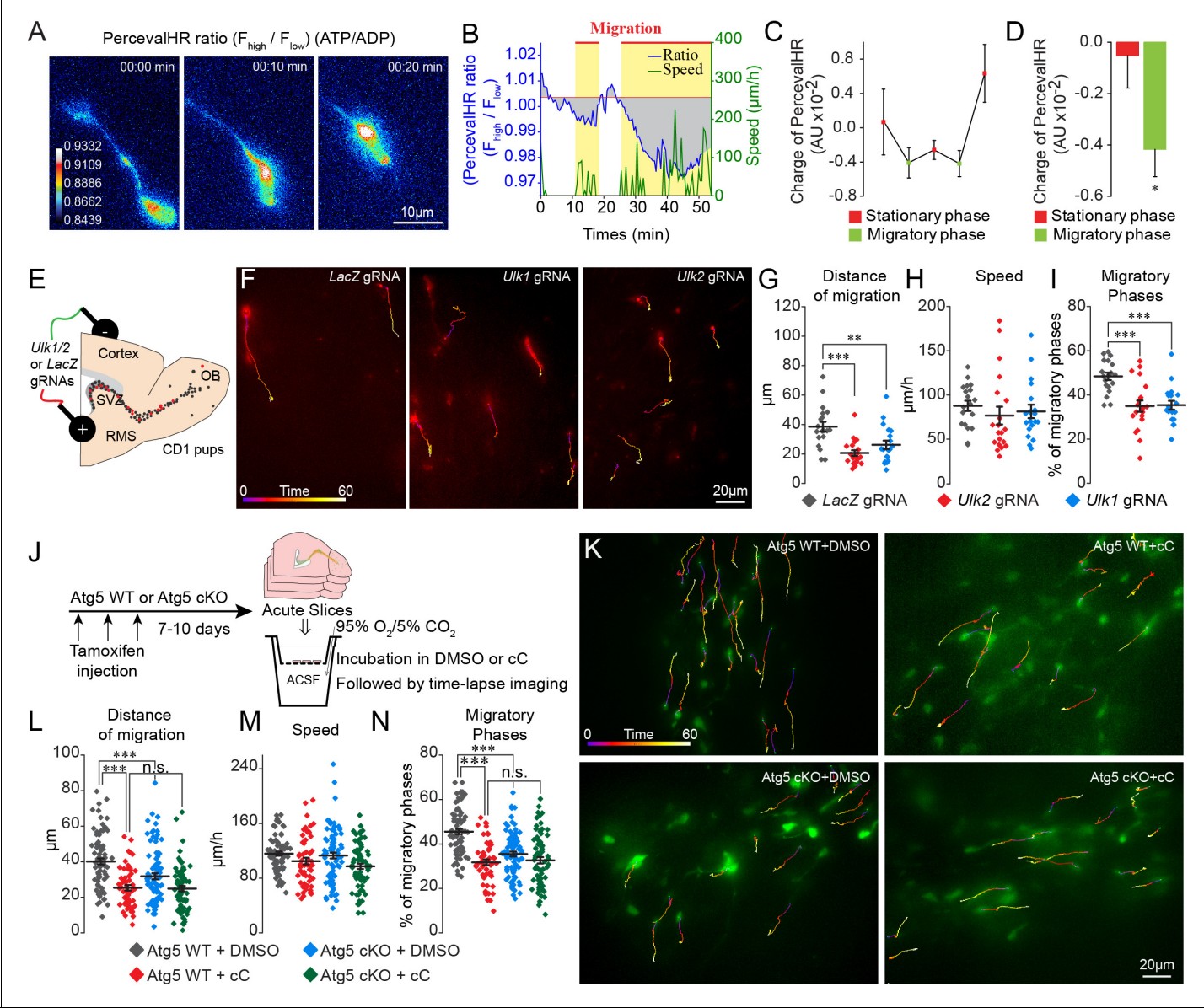

**Figure 4.** ATP/ADP ratios are dynamically modulated during cell migration and trigger autophagy via AMPK. (**A–B:**) Example and quantification of changes in the ATP/ADP ratios of individual neuroblasts during the different cell migration phases. Yellow boxes indicate the migratory phases. As the duration of the migratory and stationary phases in individual neuroblasts as well as across different cells varies, we calculated the changes in the ATP/ADP ratio as the changes in the charge (area under the curve; gray areas) divided by the duration of the different phases. The baseline for the area under the curve was fixed as the mean PercevalHR ratio during the stationary period. (**C–D:**) Quantification of changes in the ATP/ADP ratio of neuroblasts during the migratory and stationary phases (n = 19 cells from eight animals). Data are expressed as means ± SEM, *p<0.05 with the Student t-test. (**E:**) Experimental procedure for the electroporation of plasmids expressing Cas9-T2A-mCherry and gRNAs. (**F:**) Time-lapse imaging of neuroblasts electroporated with *LacZ*, *Ulk1*, and *Ulk2* gRNAs in acute brain sections. (**G–I:**) Distance of migration, speed of migration, and percentage of migratory phases of cells electroporated with *LacZ*, *Ulk1*, and *Ulk2* gRNAs (n = 19 cells for *LacZ* and *Ulk1* gRNAs, and 20 cells for *Ulk2* gRNAs from 5, 7, and four animals, respectively, **p<0.005 and ***p<0.001 with a one-way ANOVA followed by an LSD-Fisher post hoc test). (**J:**) Experiments to study the role of AMPK in autophagy-dependent neuronal migration. Acute sections from Atg5 WT and Atg5 cKO mice were incubated for 2 hr with DMSO or cC (20 µM) followed by time-lapse imaging of cell migration in the presence of DMSO or cC. (**K:**) Example of time-lapse imaging of cell migration of Atg5 WT and Atg5 cKO neuroblasts in the presence of DMSO or cC. (**L–N:**) Distance of migration, speed of migration, and percentage of migratory phases of neuroblasts (n = 79 cells from five mice for Atg5 WT+DMSO, n = 56 cells from five mice for Atg5 WT+cC, n = 85 cells from six mice for Atg5 cKO+DMSO, and n = 72 cells from seven mice for Atg5 cKO+cC, ***p<0.001 with a one-way ANOVA followed by an LSD-Fisher post hoc test). Individual values and means ± SEM for all the time-lapse imaging experiments are shown. See also *Video 4*.

The online version of this article includes the following figure supplement(s) for figure 4:

*Figure 4 continued on next page*

**Figure supplement 1.** Validation of the PercevalHR functionality in migrating neuroblasts.
**Figure supplement 2.** Validation of gRNA efficiency by high-resolution melting (HRM) PCR.

for Atg5 WT and n = 31 cells from three mice for Atg5 cKO, *Figure 5C–D*). We obtained similar results with cells electroporated with *Atg12* gRNA. Immunolabeling for paxillin in brain sections from *Atg12* gRNA-electroporated mice revealed that this focal adhesion molecule accumulates in the leading process of neuroblasts compared to control *LacZ* gRNA-electroporated cells (n = 10 cells from three mice for *LacZ* gRNA and for *Atg12* gRNA, *Figure 5E–F*). Altogether these data indicate that autophagy regulates neuronal migration in vivo by recycling paxillin, which preferentially localizes at the leading edge of migrating neuroblasts.

## The pharmacological modulation of cell migration induces changes in autophagy

Although our results show that pharmacological and genetic perturbations of autophagy-related genes affect cell migration and that energy levels are linked to autophagy induction, it remained unclear whether autophagy levels are dynamically regulated in response to migration-promoting or migration-inhibiting cues to sustain cell migration. To address this issue, we incubated adult forebrain sections with factors known to affect neuroblast migration in the RMS (*Snapyan et al., 2009*; *Bolteus and Bordey, 2004*; *Lee et al., 2006*; *Shinohara et al., 2012*), dissected out the RMS, and immunoblotted these samples for LC3 (*Figure 6*). We used BDNF (10 ng/mL), which promotes neuroblast migration (*Snapyan et al., 2009*), GABA (10 μM), which reduces the speed of migration (*Snapyan et al., 2009*; *Bolteus and Bordey, 2004*), GM60001 (100 μM), which inhibits matrix metalloproteinases and impacts neuroblast migration in the RMS (*Lee et al., 2006*; *Rempe et al., 2018*), and Y27632 (50 μM) and blebbistatin (100 μM), which affect neuroblast migration by inhibiting Rock and myosin II (*Shinohara et al., 2012*). Intriguingly, all these treatments induced changes in LC3-II, the lipidated autophagosomal form of LC3 (*Figure 6A and D*), suggesting that changes in autophagy are a common signature for neurons displaying either up- or down-regulated migration. Since an increase in LC3-II levels could reflect either an increase in autophagic flux (i.e. an increased autophagosome formation) or an accumulation of autophagosomes (i.e. an impairment of autophagosome fusion with lysosomes) (*Klionsky et al., 2012*), we investigated proteins that may be sequestered and recycled by the autophagic process, such as p62/SQSTM1, a major autophagy substrate (*Pankiv et al., 2007*), or paxillin (*Sharifi et al., 2016*; *Kenific et al., 2016*). We thus performed immunoblotting for p62 and paxillin on RMS samples. Interestingly, while the level of p62 did not change after these pharmacological treatments (*Figure 6B*), the level of paxillin increased when cell migration decreased and decreased when cell migration increased (*Figure 6A–E*). This is in line with our previous analysis showing the presence of paxillin-GFP in RFP+ autophagosomes and the accumulation of paxillin in Atg5- and Atg12-deficient neuroblasts (*Figure 5*). To provide further support for dysregulated autophagy in the context of affected cell migration, we immunostained sections containing neuroblasts infected with RFP-GFP-LC3 for Lamp1, a lysosome marker, to determine the autophagosome/autolysosome ratio (*Figure 6F–G*). In line with previous results, drugs that decreased migration caused an increase in the autophagosome/autolysosome ratio, indicating deficient autophagy (*Figure 6G*). These results indicate that several signaling pathways known to affect neuroblasts

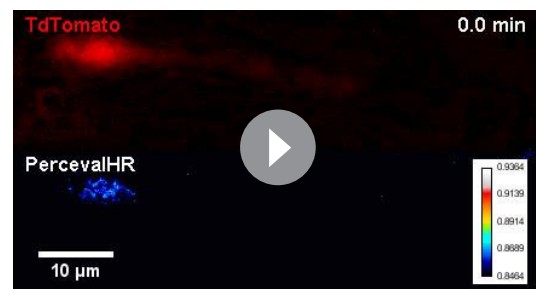

**Video 4.** The ATP/ADP ratio is dynamically modulated during cell migration. Time-lapse imaging of neuroblasts infected with PercevalHR-encoding and Td-Tomato-encoding lentiviruses. PercevalHR was used for ratiometric measurements of changes in the ATP/ADP ratio. The ATP/ADP ratio is shown. Note that the ATP/ADP ratio dynamically changes during the different cell migration phases. The time is indicated in the upper left corner.
https://elifesciences.org/articles/56006#video4

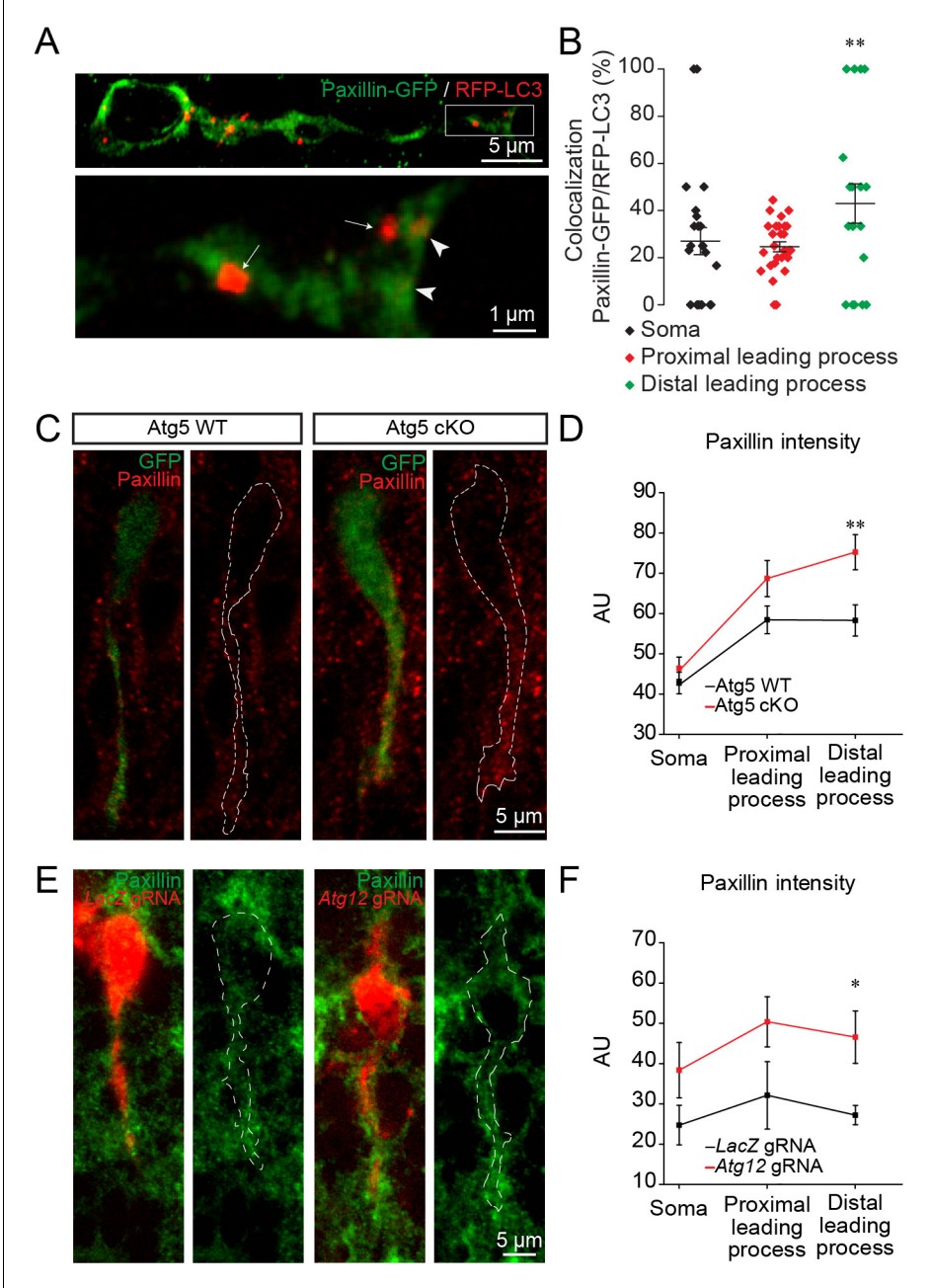

**Figure 5.** Autophagy affects neuronal migration by the recycling of paxillin. (**A**) Example of paxillin-GFP and RFP-LC3 punctae in migrating neuroblasts. Arrowheads indicate paxillin-GFP/RFP-LC3 punctae whereas arrows indicate RFP-LC3 punctae. (**B**) Quantification of paxillin-GFP+ and RFP-LC3+ vesicles in the cell body and the proximal and distal parts of the leading process (28.4 ± 1.7% of GFP+/RFP+ in the cell body, 24.5 ± 2.0% in the proximal leading process, and 42.5 ± 7.9% in the distal leading process, n = 30 cells from three mice; **p<0.005 with a one-way ANOVA followed by an LSD-Fisher post hoc test). (**C**) Example of paxillin immunolabeling in neuroblasts from Atg5 WT and Atg5 cKO mice. (**D**) Quantification of the fluorescence intensity of paxillin in the soma and in the proximal and distal parts of the leading process. For Atg5 WT mice, the fluorescence was 47.7 ± 2.7 AU for the soma, 58.43 ± 3.4 AU for the proximal leading process, and 58.3 ± 3.9 AU for the distal part of the leading process; for Atg5 cKO mice, the fluorescence intensity was 46.4 ± 4.62 AU for the soma, 76.3 ± 4.5 AU for the proximal leading process, and 74.2 ± 4.8 AU for the distal part of the leading process; n = 38 cells from three mice for Atg5 WT and n = 31 cells from three mice for Atg5 cKO (**p<0.005 with Student t-test). (**E**) Example of paxillin immunolabeling in neuroblasts expressing *LacZ* gRNA or *Atg12* gRNA. (**F**) For the *LacZ* gRNA, the fluorescence intensity was 24.7 ± 4.9 AU for the soma, 32.1 ± 8.4 AU for the proximal part of the leading process, and 27.2 ± 2.4

*Figure 5 continued*

AU for the distal part of the leading process; for the *Atg12* gRNA, the fluorescence intensity was 38.4 ± 1.6 AU for the soma, 50.4 ± 5.5 AU for the proximal part of the leading process, and 46.5 ± 2.5 AU for the distal part of the leading process; n = 10 cells from three mice for *LacZ* gRNA and for *Atg12* gRNA, *p<0.05 with the Student t-test.

migration converge on autophagy to sustain the pace and periodicity of cell migration through a constant assembly and disassembly of focal adhesions required for cell motility.

## Discussion

We show that cell migration in both the developing and adult RMS is sustained by a dynamic interplay between ATP/ADP levels and autophagy, which concomitantly regulates the pace and periodicity of the migratory and stationary phases. Pharmacological and genetic impairments of autophagy and AMPK both reduced cell migration by prolonging the stationary phases. Autophagy was also dynamically regulated in response to migration-promoting or migration-inhibiting factors and was required for the recycling of the focal adhesion molecule paxillin.

Tracking intracellular ATP/ADP levels in migrating cells allowed us to demonstrate that the efficiency of cell migration and the periodicity of the migratory and stationary phases are directly linked to the energy status of the cells. The migratory phase led to a drop in ATP/ADP levels that, in turn, was associated with the entry of the cells into the stationary phase. During the stationary phases, ATP/ADP levels recovered due to AMPK activation, following which the cells re-entered the migratory phase. Although it is known that cell migration entails a considerable bioenergetic demand (*van Horssen et al., 2009*; *O'Connell et al., 2014*; *Zhou et al., 2014*), it is

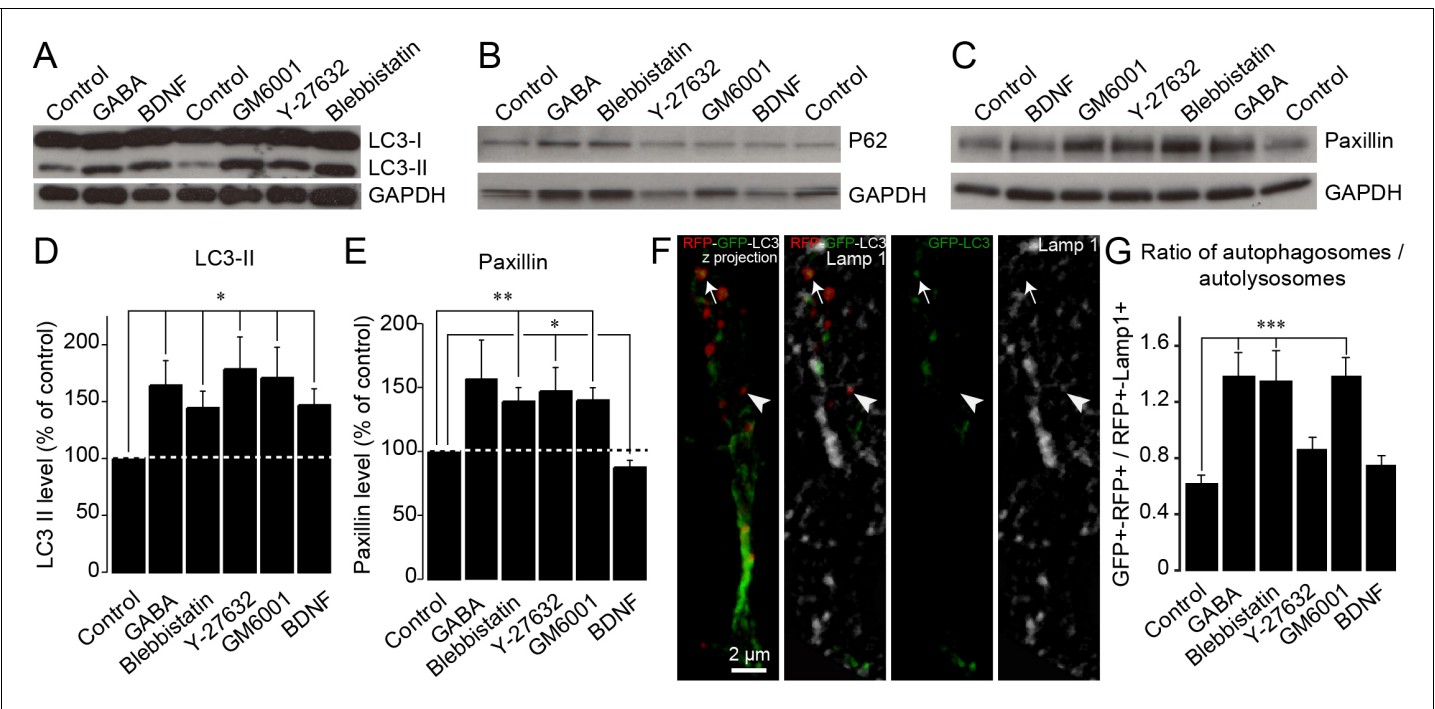

**Figure 6.** Autophagy is dynamically regulated by migration-promoting and migration-inhibiting cues and is required for the recycling of paxillin. (**A–C:**) Immunoblotting for the lipidated form of LC3 (LC3-II), p62, and paxillin on RMS samples dissected from acute sections previously incubated with BDNF, GABA, GM60001, Y27632, or blebbistatin for 2 hr. GAPDH was used as a housekeeping protein. (**D–E:**) Quantification of LC3-II and paxillin levels after the pharmacological manipulation of cell migration (n = 5–7 mice for both groups, *p<0.05 and **p<0.005 with a Student t-test). (**F**): Example of a cell infected with a retrovirus expressing the LC3-GFP-RFP fusion protein and immunostained for Lamp1 to label autophagosomes (GFP+/RFP+) and autolysosomes (RFP+/Lamp1+). (**G:**) Percentage of autophagosomes and autolysosomes after a 2 hr incubation with BDNF, GABA, GM60001, Y27632, or blebbistatin. The autophagosome/autolysosome ratio was assessed for each cell, and the results are expressed as means ± SEM. ***p<0.001 with a one-way ANOVA followed by a post hoc LSD-Fisher test.

unclear whether and how energy consumption is dynamically regulated during cell migration. Zhang et al. used intracellular measurements of ATP/ADP levels during breast cancer cell invasion to show that energy levels play a role in the coordinated leader-follower cell transition during collective migration (*Zhang et al., 2019*). ATP/ADP levels are also proportional to the propensity of cancer cells to migrate in vitro (*Zanotelli et al., 2018*). Our results further strengthened the link between cell migration and ATP/ADP levels and showed that dynamic changes in ATP/ADP levels are required for sustaining cell migration by regulating the duration and periodicity of the migratory and stationary phases.

We also showed that a decrease in ATP/ADP levels leads to the activation of AMPK, which in turn triggers autophagy. One of the downstream targets of AMPK is Ulk1/2 (*Egan et al., 2011*), and the Ulk1/FIP2000 complex is a key activator of autophagy (*Jung et al., 2009*; *Hara et al., 2008*). This may provide a mechanistic link between energy consumption and autophagy induction during cell migration. Indeed, the genetic impairment of *Ulk1*, *Ulk2*, and other autophagy genes such as *Atg5* and *Atg12* in neuroblasts resulted in impaired cell migration, while the inhibition of AMPK in Atg5-deficient cells failed to induce any additional changes in cell migration. Our CRISPR-Cas9 gene editing in pups and viral vector and transgenic mice approaches in adults demonstrated that autophagy regulates cell migration in both the developing and the adult RMS. The present study thus adds to the growing body of evidence indicating that autophagy plays a major role in cell migration. In line with our data are observations showing that changes in autophagy are accompanied by modifications in neuronal migration (*Gstrein et al., 2018*; *Peng et al., 2012*; *Petri et al., 2017*; *Li et al., 2019*). However, the modifications in autophagy observed in these studies resulted from genetic manipulations of Vps15, Vps18, let-7, and Foxp1 that affected not only autophagy but also other cellular processes that may have impacted neuronal migration (*Gstrein et al., 2018*; *Peng et al., 2012*; *Petri et al., 2017*; *Li et al., 2019*). For example, mutations in Vps15 and Vps18, in addition to autophagy defects, also perturb other vesicle transport pathways such as endocytosis (*Peng et al., 2012*; *Gstrein et al., 2018*), which regulates neuronal migration (*Cosker and Segal, 2014*; *Yap and Winckler, 2012*). Genetic manipulations of autophagy-related genes have also been performed and led to changes in migration (*Kenific et al., 2016*; *Sharifi et al., 2016*; *Tuloup-Minguez et al., 2013*; *Li et al., 2015*). These studies were, however, performed in vitro using non-neuronal cells and resulted in contradictory results (*Kenific et al., 2016*; *Sharifi et al., 2016*; *Tuloup-Minguez et al., 2013*; *Li et al., 2015*). shRNA against *Atg7* or an *Atg5* knock-out increases the migration of HeLa cells and murine embryonic fibroblasts (*Tuloup-Minguez et al., 2013*) and endothelial progenitor cells (*Li et al., 2015*). In contrast, autophagy promotes cell migration by recycling focal adhesions while the genetic impairment of autophagy leads to decreased cell migration (*Kenific et al., 2016*; *Sharifi et al., 2016*). As autophagy depends on the cellular context (*Klionsky et al., 2012*), it is conceivable that differences between these studies can be explained by the types of cells studied and the cellular context in which they were studied. Our ex vivo and in vivo results showed that autophagy promotes the migration of developing neurons. Furthermore, we showed that autophagy is very dynamic during cell migration, is dynamically regulated in response to migration-promoting and migration-inhibiting factors, and is involved in the recycling of focal adhesions. These results are in agreement with those of previous in vitro studies showing that LC3-II directly targets the focal adhesion protein paxillin (*Sharifi et al., 2016*; *Kenific et al., 2016*). Changes in autophagy levels in response to migration-promoting or migration-inhibiting factors are required to cope with the capacity of a cell to migrate by recycling focal adhesions. Indeed, our results suggest that decreases in cell migration are linked to a lower recycling rate of paxillin and a higher autophagosome/autolysosome ratio, while the promotion of cell migration tends to increase paxillin turnover. During cell migration, autophagy not only recycles focal adhesion proteins but also inhibits focal adhesion kinase through the Atg13/Ulk1/FIP2000 complex, which promotes cell immobility (*Caino et al., 2013*). Autophagy activator protein Vps15, which is part of Vps34-PI3-kinase complex I, also mediates the stability of the cytoskeleton proteins actin and tubulin by decreasing Pak1 activity (*Gstrein et al., 2018*). On the other hand, autophagy is involved in the secretion of matrix metalloproteases (MMPs) and pro-migratory cytokine interleukin-6 (IL6), which promote tumor cell migration (*Lock et al., 2014*). These results suggest that autophagy may play a dual role in controlling cell migration. First, it mediates cytoskeleton stability, which maintains the stationary phase, which is required for cellular homeostasis and the regeneration of ATP/ADP levels. Second, it mediates the recycling of focal adhesions and the release of pro-migratory factors, which are required for the

entry of cells into the migratory phase. Altogether, our results showed that the dynamic interplay between energy levels and autophagy in migrating cells is required to sustain neuronal migration and the periodicity of the migratory and stationary phases.

# Materials and methods

Key resources table

| Reagent type (species) or resource | Designation | Source or reference | Identifiers | Additional information |
|---|---|---|---|---|
| Strain, strain background (*M. musculus*, male and female) | CAG-CAT-EGFP | *Waclaw et al., 2010* | | |
| Strain, strain background (*M. musculus*, male and female) | *Slc1a3(Glast)*<sup>CreErt2</sup> mice | *Mori et al., 2006* | | |
| Strain, strain background (*M. musculus*, male and female) | *Atg5*<sup>fl/fl</sup> | Riken | B6.129S-*Atg5* < tm1Myok> RRID:IMSR_RBRC02975 | |
| Strain, strain background (*M. musculus*, male and female) | CD1 | Charles Rivers | Strain code: 022 | |
| Strain, strain background (*M. musculus*, male) | C57BL/6NCRL | Charles Rivers | Strain code: 027 | |
| Strain, strain background (*M. musculus*, male and female) | GFAP-GFP | Jackson | Strain code: 003257FVB/N-Tg (GFAPGFP)14Mes/J | |
| Transfected construct (Synthetic) | RV-GFP | Molecular Tool Platform, CERVO Brain Research Center | Lot #RV 39 | Retroviral construct |
| Transfected construct (Synthetic) | RV-Cre-mNeptune | Molecular Tool Platform, CERVO Brain Research Center | Lot #RV 34 | Retroviral construct |
| Transfected construct (*R. norvegicus*) | RV- RFP-GFP-LC3 | Molecular Tool Platform, CERVO Brain Research Center | Lot #RV15 | Retro viral construct |
| Transfected construct (Synthetic) | LV-CMV-PercevalHR | University of North Carolina Vector Core Facility | Lot #43–44213 PercevalHR | Lentiviral construct |
| Transfected construct (Synthetic) | LV-CMV-TdTomato | University of North Carolina Vector Core Facility | Lot #43–44213 TdTomato | Lentiviral construct |
| Antibody | Anti-GFP (Rabbit polyclonal) | Thermo Fisher Scientific | Cat#A-11122; RRID:AB_221569 | DAB (1:1000) |
| Antibody | Anti-GFP (Chicken polyclonal) | Avés | GFP-1020; RRID:AB_10000240 | IF (1:1000) |
| Antibody | Anti-LC3B (Rabbit polyclonal) | Novus | Cat#NB100-2220, RRID:AB_10003146 | IF (1:200) WB (1:1000) |
| Antibody | Anti- Cleaved LC3A (Rabbit polyclonal) | Abgent | Cat#AP1805a, RRID:AB_2137587 | IF (1:100) |
| Antibody | Anti-murine Atg5 (Rabbit polyclonal) | Novus | Cat#NB110-53818, RRID:AB_828587 | IF (1:400) |
| Antibody | Anti-LAMP-1 (CD107a) (Rabbit polyclonal) | Millipore | Cat#AB2971, RRID:AB_10807184 | IF (1:500) |

*Continued on next page*

*Continued*

| Reagent type (species) or resource | Designation | Source or reference | Identifiers | Additional information |
|---|---|---|---|---|
| Antibody | Anti-paxillin (Mouse monoclonal) | BD Biosciences | Cat#610051, RRID:AB_397463 | IF (1:100) WB (1:1000) |
| Antibody | Anti-P62/SQSTM1 (Rabbit polyclonal) | Proteintech | Cat#18420–1-AP, RRID:AB_10694431 | WB (1:500) |
| Antibody | Anti-GAPDH (Mouse monoclonal) | Thermo Fisher Scientific | Cat#MA5-15738, RRID:AB_10977387 | WB (1:5000) |
| Antibody | Anti-rabbit IgG, biotin-SP conjugate (Goat polyclonal) | Millipore | Cat#AP132B, RRID:AB_11212148 | DAB (1:1000) |
| Antibody | Anti-GFP (Rabbit polyclonal) | Abcam | Cat#ab290, RRID:AB_303395 | EM (1:1000) |
| Antibody | Nanogold-Fab goat anti-rabbit | Nanoprobe | Cat#2004, RRID:AB_2631182 | EM (1:20) |
| Antibody | Anti-Atg12 (Rabbit polyclonal) | Abcam | Cat#ab155589 | IF (1:500) |
| Antibody | Anti-mCherry (16D7) (Rat monoclonal) | Thermo Fisher Scientific | Cat #M11217, RRID:AB_2536611 | IF (1:1000) |
| Recombinant DNA reagent | FUGW-PercevalHR (plasmid) | Addgene | RRID:Addgene_49083 | |
| Recombinant DNA reagent | pCSCMV:tdTomato (plasmid) | Addgene | RRID:Addgene_30530 | |
| Recombinant DNA reagent | ptfLC3 (plasmid) | Addgene | RRID:Addgene_21074 | |
| Recombinant DNA reagent | PU6-BbsI-CBh-Cas9-T2AmCherry (plasmid) | Addgene | RRID:Addgene_64324 | |
| Recombinant DNA reagent | pmRFP-LC3 (plasmid) | Addgene | RRID:Addgene_21075 | |
| Recombinant DNA reagent | pRK GFP paxillin (plasmid) | Addgene | RRID:Addgene_50529 | |
| Recombinant DNA reagent | GW1-pHRed (plasmid) | Addgene | RRID:Addgene_31473 | |
| Sequence-based reagent | *Atg12*-gRNA1-Fw | This paper | PCR primer | CGGAAACAGCCACCCCAGAG |
| Sequence-based reagent | *Atg12*-gRNA1-Rs | This paper | PCR primer | GCCCACTAACGGATG TTGACATTACTT |
| Sequence-based reagent | *Atg12*-gRNA2-Fw | This paper | PCR primer | ACGCTGCTACGTCACTTCC |
| Sequence-based reagent | *Atg12*-gRNA2-Rs | This paper | PCR primer | GCTCTGGAAGGCTCTCGC |
| Sequence-based reagent | *Ulk1*-gRNA1-Fw | This paper | PCR primer | TCGCAAGGACCTGATTGGAC |
| Sequence-based reagent | *Ulk1*-gRNA1-Rs | This paper | PCR primer | CCTCGCAATCCCGGACTC |
| Sequence-based reagent | *Ulk1*-gRNA2-Fw | This paper | PCR primer | CATCTGCTTTTTATCCCAGCA |
| Sequence-based reagent | *Ulk1*-gRNA2-Rs | This paper | PCR primer | CTGCAACAGAGCCAGGAG |
| Sequence-based reagent | *Ulk2*-gRNA1-Fw | This paper | PCR primer | TACTGCAAGCGGGACCT |
| Sequence-based reagent | *Ulk2*-gRNA1-Rs | This paper | PCR primer | TTTCGCACCAGACAACGGG |
| Sequence-based reagent | *Ulk2*-gRNA2-Fw | This paper | PCR primer | CTCTGAGTGAAGAT ACTATCAGAGTG |

*Continued on next page*

Continued

| Reagent type (species) or resource | Designation | Source or reference | Identifiers | Additional information |
|---|---|---|---|---|
| Sequence-based reagent | *Ulk2*-gRNA2-Rs | This paper | PCR primer | GATCCCTGTGGATTATCCCTTT |
| Commercial assay or kit | DNeasy Blood and Tissue Kit | Qiagen | Cat#69504 | |
| Commercial assay or kit | LightCycler 480 High Resolution Melting Master | Roche | Cat#04909631001 | |
| Commercial assay or kit | VECTASTAIN ABC-Peroxidase Kit | Vector Laboratories | Cat#PK4000 | |
| Commercial assay or kit | HQ Silver Enhancement kit | Product | Cat#2012 | |
| Chemical compound, drug | NeuroCult Proliferation Supplement | StemCell Technologies | Cat#05701 | |
| Chemical compound, drug | NeuroCult Basal Medium | StemCell Technologies | Cat#05700 | |
| Chemical compound, drug | EGF | Sigma-Aldrich | Cat#E4127 | |
| Chemical compound, drug | bFGF | Sigma-Aldrich | Cat#SRP4038-50UG | |
| Chemical compound, drug | Heparin | StemCell Technologies | Cat#07980 | |
| Chemical compound, drug | MgCl$_2$ | Sigma-Aldrich | Cat#M8266; CAS 7786-30-3 | |
| Chemical compound, drug | Paraformaldehyde | Sigma-Aldrich | Cat#P6148; CAS: 30525-89-4 | |
| Chemical compound, drug | Tamoxifen | Sigma-Aldrich | Cat#T5648; CAS: 10540-29-1 | |
| Chemical compound, drug | Anhydrous ethyl alcohol | Commercial Alcohols | 1019C | |
| Chemical compound, drug | Sunflower seed oil | Sigma-Aldrich | Cat#S5007 CAS: 8001-21-6 | |
| Chemical compound, drug | Protease Inhibitor Cocktail Set III | Millipore | 539134 | |
| Chemical compound, drug | Blebbistatin | Research Chemicals Inc Toronto | Cat#B592490-10 CAS: 674289-55-5 | |
| Chemical compound, drug | GM6001 | Abmole | Cat#M2147-10MG CAS: 142880-36-2 | |
| Chemical compound, drug | Y-27632 | Cayman Chemical | Cat#10005583–5 CAS: 129830-38-2 | |
| Chemical compound, drug | Compound C | EMD Millipore | Cat#171260–10 MG CAS: 866405-64-3 | |
| Peptide, recombinant protein | BDNF | Peprotech | Cat#450–02 | |
| Peptide, recombinant protein | GABA | Sigma-Aldrich | Cat#A5835-25G CAS: 56-12-2 | |
| Software, algorithm | Imaris 7.2 | Bitplane, Oxford Instrument | https://imaris.oxinst.com/ | |
| Software, algorithm | MATLAB 2016a | Mathworks | https://www.mathworks.com/ | |

*Continued on next page*

Continued

| Reagent type (species) or resource | Designation | Source or reference | Identifiers | Additional information |
|---|---|---|---|---|
| Software, algorithm | MATLAB scripts PercevalHR imaging analysis | This paper | https://github.com/SagLab-CERVO/PercevalHR-fluorescence-intensity | To track cell and measure PercevalHR fluorescence intensity (*Labrecque et al., 2020*; copy archived at https://github.com/elifesciences-publications/PercevalHR-fluorescence-intensity) |
| Software, algorithm | ImageJ | NIH | https://imagej.net/Welcome | |
| Software, algorithm | Origin 2016 | OriginLab Corporation | https://www.originlab.com/ | |
| Software, algorithm | Statistica 6.1 | Stat Soft | N/A | |
| Software, algorithm | Primer-Blast software | NCBI | https://www-ncbi-nlm-nih-gov.acces.bibl.ulaval.ca/tools/primer-blast/ | |
| Software, algorithm | ChopChop online software | *Labun et al., 2019* | https://chopchop.cbu.uib.no/ | |
| Software, algorithm | LightCycler 480 SW1.5.1 software | Roche | 04994884001 | |

## Animals

Experiments were performed using two- to four-month-old C57BL/6 mice (Charles River, strain code: 027), *glial fibrillary acidic protein* (*GFAP*)-GFP mice (The Jackson Laboratory, strain code: 003257 FVB/N-Tg(*GFAP*GFP)14Mes/J), *Slc1a3*(*Glast*)$^{CreErt2}$::*Atg5*$^{fl/fl}$::CAG-CAT-GFP (Atg5 cKO, strain code: B6.129S-*Atg5* < tm1Myok>, RRID:IMSR_RBRC02975), and *Slc1a3*(*Glast*)$^{CreErt2}$::*Atg5*$^{wt/wt}$::CAG-CAT-GFP (Atg5 WT) mice as well as two-week-old CD-1 pups (The Jackson Laboratory, strain code: 022), which were electroporated on postnatal day 1–2. To obtain the Atg5 WT and Atg5 cKO mice, we first crossed *Slc1a3*(*Glast*)$^{CreErt2}$ mice (*Mori et al., 2006*) with CAG-CAT-GFP mice *Waclaw et al., 2010* to obtain *Slc1a3*(*Glast*)$^{CreErt2}$::GFP mice. This mouse strain was then crossed with *Atg5*$^{fl/fl}$ mice (B6.129S-*Atg5* < tm1Myok>, RBRC02975, Riken) to obtain the Atg5 WT and Atg5 cKO mice. All the experiments were approved by the Université Laval animal protection committee. The mice were housed one to five per cage. They were kept on a 12 hr light/dark cycle at a constant temperature (22°C) with food and water ad libitum.

## Stereotactic injections

For the retrovirus and lentivirus injections, C57BL/6 mice were anesthetized with isoflurane (2–2.5% isoflurane, 1 L/min of oxygen) and were kept on a heating pad during the entire surgical procedure. Lentiviruses or retroviruses were stereotactically injected in the dorsal and ventral subventricular zones (SVZd and SVZv) at the following coordinates for C57BL/6 (with respect to the bregma): anterior-posterior (AP) 0.70 mm, medio-lateral (ML) 1.20 mm, dorso-ventral (DV) 1.90 mm, and AP 0.90 mm, ML 1 mm, and DV 2.75 mm. For *Atg5*$^{fl/fl}$ and *Atg5*$^{wt/wt}$, the following coordinates were used: AP 0.90 mm and AP 1.10 mm were used for SVZd and SVZv, respectively. The following viruses were used: CMV-PercevalHR (1 × 10$^{10}$ TU/mL, produced at the University of North Carolina Vector Core Facility based on plasmid # 49083, Addgene, kindly provided by Dr. Yellen, Harvard Medical School); CMV-Td-Tomato-encoding lentivirus (1.5 × 10$^{10}$ TU/mL, produced at the University of North Carolina Vector Core Facility based on plasmid # 30530, Addgene, kindly provided by Dr. Ryffel, University of Duisburg-Essen); RFP-GFP-LC3-encoding retrovirus (9.3 × 10$^7$ TU/mL, produced at the Molecular Tools Platform at the CERVO Brain Research Center based on plasmid #21074, Addgene, kindly provided by Dr. Yoshimori, Osaka University); GFP-encoding retrovirus (2.9 × 10$^7$ TU/mL, Molecular Tools Platform at the CERVO Brain Research Center), and Cre-mNeptune-encoding retrovirus (4 × 10$^7$ TU/mL, Molecular Tools Platform at the CERVO Brain Research Center).

For the electroporation of plasmids, P1-P2 CD-1 pups were anesthetized using isoflurane (2–2.5% isoflurane, 1 L/min of oxygen). The plasmids (1.3 µL, 3–6 µg/µL total) were injected in the lateral ventricle at the following coordinates (with respect to the lambda): AP 1.8 mm, ML 0.8 mm, and DV 1.6 mm. Immediately after the injection, an electric field (five 50 ms pulses at 100 mV at 950 ms intervals) was applied using an electrode positioned on the surface of the bones. The pups were used 8 to 15 days post-injection. We used gRNAs for *Ulk1*, *Ulk2*, and *Atg12*. We used gRNAs for *LacZ* as a control. We used two different gRNAs for each gene and co-electroporated them. To assess the autophagy-dependent recycling of paxillin, we co-electroporated pmRFP-LC3 (Addgene, #21075, kindly provided by Dr. Yoshimori, Osaka University) with pRK paxillin-GFP (Addgene, #50529, kindly provided by Dr. Yamada, National Institute of Dental and Craniofacial Research) at P1-P2 and analyzed migrating neuroblasts in the RMS 8–10 days later. To assess changes in the pH of migrating cells, we co-electroporated pHRed (Addgene, #31473, kindly provided by Dr. Yellen, Harvard Medical School) with FUGW-PercevalHR (Addgene, #49083, kindly provided by Dr. Yellen, Harvard Medical School).

## CRISPR target site selection and assembly

The gRNAs were designed and selected using ChopChop online software (*Labun et al., 2019*). We used two different gRNAs to target each gene to increase the efficiency of the CRISPR editing. The gRNAs were cloned in the BbsI site of the PU6-(BbsI)_CBh-Cas9-T2A-mCherry plasmid (Addgene, #64324, kindly provided by Dr. Kuehn, Berlin Institute of Health). The following gRNA sequences were used:

| gRNA sequences | (5′ −3′) |
| --- | --- |
| *Atg12* gRNA1 | TGCAGTTTCGCCCGGAACGG |
| *Atg12* gRNA2 | GGTTGTGCTGCAGCTCCCCT |
| *Ulk1* gRNA1 | CTTCAAGGGTCGACACCGCG |
| *Ulk1* gRNA2 | GGGAGACATCAGCTCCCCTG |
| *Ulk2* gRNA1 | CCCCCGGAAGACCACAGCGA |
| *Ulk2* gRNA2 | AGTGTTTCTCCATCAGATTG |
| *LacZ* gRNA | TGCGAATACGCCCACGCGAT |

The efficiency of the gRNAs was verified by high-resolution melting curve PCR (*Thomas et al., 2014*) on primary cultures of adult NPCs transfected with the above-mentioned plasmids. Briefly, the thin layer of SVZ bordering the lateral ventricle, excluding the striatal parenchyma and corpus callosum, was dissected from the SVZ of 8- to 12-week-old C57BL/6 mice. The tissue was minced and digested in a 0.05% trypsin-EDTA solution, following which an equal volume of soybean trypsin inhibitor was added. After trituration, the resulting single cell suspension was cultured in NeuroCult Basal Medium (StemCell Technologies, #05701) with NeuroCult Proliferation Supplement (StemCell Technologies, #05701), EGF and bFGF (10 ng/mL each, Sigma-Aldrich, #E4127 and #SRP4038-50UG), and heparin (2 µg/mL, StemCell Technologies, #07980).

Genomic DNA was isolated using DNeasy Blood and Tissue kits (Qiagen, #69504) according to the manufacturer's protocol. Primers were designed using Primer-Blast software (https://www.ncbi.nlm.nih.gov/tools/primer-blast/).

The PCR reactions were performed with 5 µL of LightCycler 480 High Resolution Melting Master (Roche, #04909631001), 0.5 µL of each primer (10 µM), 1.2 µL of $MgCl_2$ (25 mM, Sigma-Aldrich, #M8266; CAS 7786-30-3), 2 µL of genomic DNA, and 1.3 µL of water for a total volume of 10 µL. The PCR was performed in a LightCycler 480 (Roche) using 96-well plates (Bio-Rad). The amplification started with an initial denaturation step at 95°C for 5 min, followed by 48 cycles at 95°C for 10 s, 60°C for 30 s, and 72°C for 25 s. Melting curves were generated over a 65–95°C range in 0.2°C increments and were analyzed using LightCycler 480 SW1.5.1 software. The following primers were used:

| HRM primers | (5' —3') |
|---|---|
| *Atg12*-gRNA1-Fw | CGGAAACAGCCACCCCAGAG |
| *Atg12*-gRNA1-Rs | GCCCACTAACGGATGTTGACATTACTT |
| *Atg12*-gRNA2-Fw | ACGCTGCTACGTCACTTCC |
| *Atg12*-gRNA2-Rs | GCTCTGGAAGGCTCTCGC |
| *Ulk1*-gRNA1-Fw | TCGCAAGGACCTGATTGGAC |
| *Ulk1*-gRNA1-Rs | CCTCGCAATCCCGGACTC |
| *Ulk1*-gRNA2-Fw | CATCTGCTTTTTATCCCAGCA |
| *Ulk1*-gRNA2-Rs | CTGCAACAGAGCCAGGAG |
| *Ulk2*-gRNA1-Fw | TACTGCAAGCGGGACCT |
| *Ulk2*-gRNA1-Rs | TTTCGCACCAGACAACGGG |
| *Ulk2*-gRNA2-Fw | CTCTGAGTGAAGATACTATCAGAGTG |
| *Ulk2*-gRNA2-Rs | GATCCCTGTGGATTATCCCTTT |

## Tamoxifen injection

For time-lapse imaging of neuroblasts in acute brain sections from Atg5 WT and Atg5 cKO mice, the tamoxifen was injected intraperitonially (180 mg/kg, Sigma-Aldrich, #T5648, CAS: 10540-29-1) once a day for 3 days. The tamoxifen was diluted in sunflower seed oil (Sigma-Aldrich, #S5007, CAS: 8001-21-6) and anhydrous ethanol (10% final, Commercial Alcohols, #1019C). The mice were used 7 to 15 days after the last tamoxifen injection.

The mice received a single intraperitoneal tamoxifen injection (180 mg/kg, Sigma-Aldrich) to assess neuroblast distribution in vivo along the SVZ-OB pathway. They were sacrificed 7 days later.

## Immunostaining

The mice were deeply anesthetized with sodium pentobarbital (12 mg/mL; 0.1 mL per 10 g of body weight) and were perfused intracardially with 0.9% NaCl followed by 4% paraformaldehyde (PFA) (Sigma-Aldrich, #P6148; CAS: 30525-89-4). Brains were collected and were kept overnight in 4% PFA. Sagittal sections (30 or 40 µm) were cut using a vibratome (Leica). For immunolabeling experiments with acute brain sections, 250-µm-thick sections were fixed in 4% PFA overnight and were pre-permeabilized with methanol and acetone (30 min each at –30°C) prior to immunostaining. The brain sections were incubated with the following primary antibodies: anti-LC3B (1:200, Novus, #NB100-2220, RRID:AB_10003146), anti-LC3A (1:100, Abgent, #AP1805a, RRID:AB_2137587), anti-Atg5 (1:400, Novus, #NB110-53818, RRID:AB_828587), anti-Atg12 (1:500, Abcam, #ab155589), anti-Lamp1 (1:500, EMD Millipore, #AB2971, RRID:AB_10807184), and anti-GFP (1:1000, Avés, #GFP-1020; RRID:AB_1000024). For anti-LC3A, LC3B, and Atg5, sections were incubated in 10 mM citrate buffer (pH 6.0) for 20 min at 80°C. The primary antibodies were diluted in either 0.2% or 0.5% Triton X-100% and 4% milk. Images were acquired using an inverted Zeiss microscope (LSM 700, AxioObserver) with a 20X objective (NA: 0.9) or a 63X oil immersion objective (NA: 1.4).

For paxillin immunolabeling experiments in adult mice, the brains were kept in a 30% sucrose solution overnight, after which 15-µm-thick cryostat sections were prepared. The sections were incubated with anti-paxillin (1:100, BD biosciences, #610051, RRID:AB_397463) and anti-GFP (1:1000, Avés) primary antibodies diluted in 0.2% Triton X-100% and 4% milk. Images were acquired using an inverted Zeiss microscope (LSM 700, AxioObserver) with a 63X oil immersion objective (NA: 1.4). For paxillin immunostaining experiments in pups, brain sections were cut using a vibratome. The sections were incubated with anti-paxillin (1:100, BD Biosciences, #610051, RRID:AB_397463) and anti-mCherry (1:1000, Thermo Fisher Scientific, #M11217, RRID:AB_2536611) primary antibodies diluted in 0.2% Triton X-100% and 4% milk. Images were acquired using an inverted Olympus microscope (FV 1000, Olympus) with a 60X oil immersion objective (NA 1.40; Olympus). For both experiments, fluorescence intensity was assessed using ImageJ after background subtraction (50-pixel rolling ball radius) using median filter (2-pixel radius) plugins. The mean fluorescence intensity was calculated for the cell soma and leading process of GFP+ neuroblasts in the RMS of Atg5 WT and Atg5 cKO

mice with 20 µm bins. The first 20 µm was considered to be the proximal leading process whereas last 20 µm was considered to be the distal leading process.

For diaminobenzidine (DAB) staining, 40 µm coronal sections separated by 240 µm were taken from the anterior tip of the OB to the end of the SVZ (approximately at the bregma) of Atg5 WT and Atg5 cKO mice. They were first incubated in 3% $H_2O_2$ and then with an anti-GFP primary antibody (1:1000, Thermo Fisher Scientific, #A-11122; RRID:AB_221569). The sections were then incubated for 3 hr at room temperature with a biotinylated anti-rabbit IgG antibody (1:1000, EMD Millipore, #AP132B, RRID:AB_11212148) followed by 1 hr with avidin–biotin complex (ABC Vector Laboratories, #PK4000) and then with 0.05% 3,3-diaminobenzidine tetrahydrochloride (DAB) and 0.027% $H_2O_2$ in 1 M Tris-HCl buffer (pH 7.6). They were dehydrated in graded ethanol baths and were mounted and coverslipped in DePeX (VWR). Whole section images were acquired using Tissue-Scope 4000 (Huron), and the cell density was determined manually using the ImageJ cell counter plugin.

To determine the density of *LacZ* gRNA- and *Atg12* gRNA-expressing cells, sagittal 40 µm sections were obtained from fixed pup brains electroporated with gRNAs using a vibratome. The brain sections containing the SVZ, RMS, and OB were stained with DAPI. Whole section images were acquired using TissueScope 4000 (Huron), and the density of mCherry+ cells (either *LacZ* or *Atg12* gRNA) was determined manually using the ImageJ cell counter plugin. DAPI counterstaining was used to define the surfaces of the SVZ, RMS, RMS$_{OB}$, and OB.

## Electron microscopy

Two weeks after the i.p. injection of tamoxifen, seven mice (4 Atg5 WT and 3 Atg5 cKO) were deeply anesthetized with a mixture of ketamine (100 mg/kg, i.p.) and xylazine (10 mg/kg, i.p.). They were transcardially perfused with 40 mL of ice-cold sodium phosphate-buffered saline (0,1 M PBS; pH 7.4) followed by 100 mL of cold 2% acroleine and 100 mL of cold 4% PFA to which 0.1% glutaraldehyde was added. The brains were excised and were post-fixed for 4 hr in 4% PFA at 4°C. They were then cut with a vibratome (model VT1200 S; Leica, Germany) into 50-µm-thick sagittal sections collected in PBS.

After a 30 min incubation in a 0.1 M sodium borohydride/PBS solution at room temperature, the sagittal sections containing the RMS were immunostained for GFP. Briefly, free-floating sections were sequentially incubated at room temperature (RT) in blocking solution containing 2% normal goat serum and 0.5% gelatin (1 hr), the same blocking solution containing a rabbit anti-GFP antibody (1:1000, Abcam, #ab290, RRID:AB_303395) (24 hr, RT), and then the same blocking solution containing a 1:20 dilution of Nanogold-Fab goat anti-rabbit antibody (Nanoprobe, #2004, RRID:AB_2631182) (24 hr, 4°C). After thoroughly rinsing the sections in 3% sodium acetate buffer (pH 7.0), the gold immunostaining was amplified using an HQ Silver Enhancement Kit (Nanoprobe, #2012) according to the manufacturer's protocol. The sections were then osmicated, dehydrated in ethanol and propylene oxide, and flat-embedded in Durcupan (Fluka). Quadrangular pieces containing RMS were cut from the flat-embedded GFP-immunostained sections. After being glued to the tip of a resin block, 50 nm sections were cut using an ultramicrotome (model EM UC7, Leica). The ultrathin sections were collected on bare 150-mesh copper grids, stained with lead citrate, and examined with a transmission electron microscope (Tecnai 12; Philips Electronic, 100 kV) equipped with an integrated digital camera (XR-41, Advanced Microscopy Techniques Corp.). Labeled cell bodies were then randomly selected, and autophagosomal vesicles were identified and measured based on careful selection criteria (*Eskelinen, 2008*).

## Time-lapse imaging

Measurement of the ATP/ADP ratio: Seven days after co-injecting lentiviral particles expressing PercevalHR or TdTomato in the SVZ, the mice were sacrificed and acute sections were prepared as described previously (*Bakhshetyan and Saghatelyan, 2015*). Briefly, the mice were anesthetized with ketamine (100 mg/kg) and xylazine (10 mg/kg) and were perfused transcardially with modified oxygenated artificial cerebrospinal fluid (ACSF) containing (in mM): 210.3 sucrose, 3 KCl, 2 CaCl$_2$.2H$_2$O, 1.3 MgCl$_2$.6H$_2$O, 26 NaHCO$_3$, 1.25 NaH$_2$PO$_4$.H$_2$O, and 20 glucose. The brains were then quickly removed, and 250-µm-thick sections were cut using a vibratome (HM 650V; Thermo Fisher Scientific). The sections were kept at 37°C in ACSF containing (in mM): 125 NaCl, 3 KCl, 2

CaCl$_2$.2H$_2$O, 1.3 MgCl$_2$.6H$_2$O, 26 NaHCO$_3$, 1.25 NaH$_2$PO$_4$.H$_2$O, and 20 glucose under oxygenation for no more than 6–8 hr. PercevalHR imaging was performed using a BX61WI (Olympus) upright microscope with a 60X water immersion objective (NA = 0.9), a CCD camera (CoolSnap HQ), and a DG-4 illumination system (Sutter Instrument) equipped with a Xenon lamp for rapid wavelength switching. The field of view was chosen to have sparse virally labeled cells. The cells were excited at 430 nm and 482 nm, and images were acquired at 525 nm for PercevalHR. The imaging was performed every 30 s for 2 hr with multiple z stacks (4–8 stacks, depending on the cell orientation, at 3 μm intervals). The images were analyzed using a custom script written in Matlab. Briefly, a maximum intensity projection of the z stack was created for each wavelength. The migration of the cells was assessed based on the morphological marker TdTomato. The means of the fluorescence intensities at 430 nm and 482 nm were extracted, and the ratio of fluorescence for these two wavelengths was calculated. The speed of migration, as well as the migratory and stationary phases, were determined as described below. To estimate changes in the ATP/ADP ratio, we measured the charge, which was defined as the area under the ATP/ADP ratio curves for each stationary and migratory phase divided by the duration of the phases. To calculate the area under the curves, we used the mean of the PercevalHR ratio during the stationary phases of the entire time-lapse movie of an individual cell as the baseline. Combined pHRed and PercevalHR imaging was performed under the same conditions. The cells were excited at 430 and 482 nm for PercevalHR and at 440 and 585 nm for pHRed. Fluorescence was collected at 525 and 624 nm for PercevalHR and pHRed, respectively. The same calculations were made for pHRed. To compare the pHRed and PercevalHR ratios both were presented as a percentage, with 100% representing the highest value for individual cells in the phases.

For the rotenone analysis, acute brain sections were incubated in DMSO (control) or in rotenone at different concentrations (1, 10, 20 μM) for 1 hr followed by the acquisition of images for PercevalHR at the two wavelengths.

Cell migration analysis: Acute sections from Atg5 WT and Atg5 cKO mice, as well as from C57BL/6 mice injected with retroviruses and CD-1 pups electroporated with plasmids, were obtained as described previously, and time-lapse imaging was performed using the same imaging system with a 40X water immersion objective (NA = 0.8) and a mercury arc lamp as the illumination source (Olympus). For the pharmacological manipulations of cell migration, the sections were incubated at 37°C in oxygenated ACSF suplemented with drugs for 1 hr. The following drugs were used: Y-27632 (50 μM, Cayman Chemical, #10005583–5, CAS: 129830-38-2), blebbistatin (100 μM, Toronto Research Chemicals, #B592490-10, CAS: 674289-55-5), GM6001 (100 μM, Abmole, #M2147-10MG, CAS: 142880-36-2), BDNF (10 nM, Peprotech, #450–02), GABA (10 μM, Tocris, #A5835-25G, CAS: 56-12-2), bafilomycin (4 μM, Cayman Chemical, #11038, CAS: 88899-55-2), and Compound C (20 μM, EMD Millipore, #A5835-25G, CAS: 56-12-2). cC and bafilomycin were also added to the ASCF during the imaging period for the time-lapse imaging of cell migration following AMPK inhibition. Images were acquired with a 40X objective every 15–30 s for 1 hr with multiple z stacks (11 stacks with 3 μm intervals). For the assessment of autophagosome dynamics in migrating cells, a 5–10 min stream acquisition with 1.4 to 4.95 frames per second was performed. The tracking analysis was performed using Imaris8 software (Biplane) to extract the overall cell displacement and vector of cell migration. We did not take into consideration cells that remained stationary during the entire imaging period, which likely resulted in an underestimation of the autophagy-dependent effects on cell migration. The distance of cell migration corresponded to the size of the vector displacement between the coordinates of the first and last position of the cell during the 1 hr of imaging. The instantaneous speed (speed between two consecutive times points) profile of each individual cell was then plotted using Origin software, and the migratory phases were determined manually based on the speed to assess the duration and periodicity of the migratory and stationary phases. The percentage of migration was calculated as the sum of the duration of all migratory events during 1 hr of imaging. The speed of migration was determined only during the migratory phases.

For the autophagosome density analysis during the migratory and stationary phases, RFP-GFP-LC3-expressing cells were imaged as described above to determine the migratory and stationary phases. Autophagosome density (RFP+ punctae) was determined at separate time points corresponding to the beginning, middle, and end of each migratory and stationary phase. The mean density of autophagosomes per migratory and stationary phase was then calculated.

To investigate autophagosome flux dynamics, acute sections from C57BL/6 mice injected with RFP-GFP-LC3 retroviruses were obtained as described above and were imaged using a two-photon

microscope (Scientifica). Images were acquired with a water immersion 20X objective (XLUMPlanFI, NA = 0.95; Olympus) every 30 s for 1 hr with multiple z stacks (10 stacks at 4 µm intervals). Insight X3 dual lasers were used to provide excitation at 940 and 1040 nm for GFP and RFP, respectively. Emission photons were collected simultaneously using two separate GaAsP PMTs. RFP+ punctae and GFP+/RFP+ punctae were counted manually using the ImageJ cell counter plugin.

### Western blotting

Acute adult mouse forebrain sections were incubated with the pharmacological compounds. The RMS was manually dissected under an inverted microscope (Olympus FV1000, 10X objective, NA = 0.4). The tissue was snap frozen in liquid nitrogen and was then incubated in lysis buffer (50 mM HCl, 1 mM EDTA, 1 mM EGTA, 1 mM sodium orthovanadate, 50 mM sodium fluoride, 5 mM sodium pyrophosphate, 10 mM sodium β-glycerophosphate, 0.1% 2-mercaptoethanol, 1% Triton X-100, pH 7.5) supplemented with a protease inhibitor cocktail III (Calbiochem, #539134). The concentration of total protein was measured using the Bradford assay (Bio-Rad). Proteins were separated on 16% NuPage gels (Invitrogen) in SDS running buffer and were transferred to nitrocellulose membranes (Life Technologies). The following primary antibodies were used: anti-LC3B (1:1000, Novus), anti-p62 (1:500, Proteintech, #18420–1-AP, RRID:AB_10694431), anti-paxillin (1:1000, BD biosciences, #610051, RRID:AB_397463), and anti-GAPDH (1:5000, Thermo Fisher Scientific, #MA5-15738, RRID:AB_10977387).

### Statistical analysis

Data are expressed as means ± SEM. Statistical significance was determined using an unpaired two-sided Student's $t$-test or a one-way ANOVA followed by an LSD-Fisher post hoc test, depending on the experiment, as indicated. Equality of variance for the unpaired t-test was verified using the F-test. The levels of significance were as follows: *$p<0.05$, **$p<0.01$, ***$p<0.001$.

## Acknowledgements

This work was supported by Canadian Institute of Health Research (CIHR) grant PJT-159733 to AS. The authors declare no conflict of interests.

## Additional information

### Funding

| Funder | Grant reference number | Author |
|---|---|---|
| Canadian Institutes of Health Research | PJT 153026 | Armen Saghatelyan |

The funders had no role in study design, data collection and interpretation, or the decision to submit the work for publication.

### Author contributions

Cedric Bressan, Conceptualization, Data curation, Formal analysis, Investigation, Writing - original draft, Writing - review and editing; Alessandra Pecora, Dave Gagnon, Marina Snapyan, Data curation, Formal analysis, Investigation; Simon Labrecque, Paul De Koninck, Software; Martin Parent, Formal analysis, Supervision; Armen Saghatelyan, Conceptualization, Data curation, Supervision, Funding acquisition, Validation, Investigation, Writing - original draft, Project administration, Writing - review and editing

### Author ORCIDs

Cedric Bressan (iD) https://orcid.org/0000-0001-8820-0354
Dave Gagnon (iD) http://orcid.org/0000-0001-7366-0665
Paul De Koninck (iD) http://orcid.org/0000-0002-6436-1062
Armen Saghatelyan (iD) https://orcid.org/0000-0003-4962-0465

## Ethics

Animal experimentation: This study was performed in strict accordance with the recommendations of Canadian Council of Animal Care. All the experiments were approved by the Université Laval animal protection committee (#2014-178 and 2019-020).

## Decision letter and Author response

Decision letter https://doi.org/10.7554/eLife.56006.sa1
Author response https://doi.org/10.7554/eLife.56006.sa2

## Additional files

### Supplementary files

- Transparent reporting form

### Data availability

All data generated or analysed during this study are included in the manuscript and supporting files.

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
