## [Decision Letter]

**Acceptance summary:**

Your work opens new avenues on the role of autophagy in the regulation of migration of adult born neurons by controlling energy metabolism.

**Decision letter after peer review:**

Thank you for submitting your article "The dynamic interplay between ATP/ADP levels and autophagy sustain neuronal migration in vivo" for consideration by *eLife*. Your article has been reviewed by two peer reviewers, and the evaluation has been overseen by a Reviewing Editor and Jonathan Cooper as the Senior Editor. The reviewers have opted to remain anonymous.

The reviewers have discussed the reviews with one another and the Reviewing Editor has drafted this decision to help you prepare a revised submission. In recognition of the fact that revisions may take longer than the two months we typically allow, until the research enterprise restarts in full, we will give authors as much time as they need to submit revised manuscripts.

Summary:

In this work you uncover the complex interplay between autophagy and energy consumption to regulate the pace and periodicity of the migratory and stationary phases in a prototypic model of migration in adult brain (the SVZ-OB). Both reviewers considered your work important as you provided evidence that 1) activity of the autophagic pathway is related to the ratio of the migratory / stationary phase, 2) activity of the autophagolysosomal pathway is related to the ATP/ADP levels and 3) autophagy targets paxillin, a focal adhesion protein that is the direct target of LC3II.

Essential revisions:

The original reviews are attached below. Most of the major points can be addressed with minimal new experiments, but may require reanalysis of data or samples you already have. Based on these reviews, it is essential to address the following key points:

1) Deepen the analysis of paxillin localization and expression.

2) Confirm the impact of Atg5 on the density of neuroblasts in the RMS (increased?) and the OB (unchanged?).

3) Confirm the loss of Atg12 protein.

4) Quantify autophagolysosomal activity (Figure 1I) by analyzing GFP+RFP/RFP punctae ratio.

In addition, while the science is strong, the novelty of the work is overstated. There is existing literature showing roles of autophagy in neuronal migration. The paper needs rewriting to accurately place your work in the context of prior research in the field.

Specific recommendations include:

i) Rewrite the Introduction and Discussion to cite the literature appropriately.

ii) The second reviewer suggests that your Results could be presented in a different way in order to make better use of your data.

---

## [Author Response]

Essential revisions:The original reviews are attached below. Most of the major points can be addressed with minimal new experiments, but may require reanalysis of data or samples you already have. Based on these reviews, it is essential to address the following key points:1) Deepen the analysis of paxillin localization and expression.

We performed additional experiments to deepen our analysis of paxillin localization and expression. First, we quantified paxillin-GFP/LC3-RFP punctae in the cell body and along the leading process of neuroblasts. Our results showed that the highest percentage of co-localization occurs in the distal portion of the leading process of neuroblasts (Figure 5A-B). We next studied how the intensity of paxillin immunolabelling (expression) is modified following perturbations in the autophagy pathway. In addition to the analysis of the intensity of paxillin immunolabelling in Atg5 WT and Atg5 cKO mice (Figure 5C-D), we also electroporated Atg12 and LacZ (as a control) gRNAs and analyzed the intensity of paxillin immunolabelling in migrating neuroblasts. Our results from both transgenic mice and CRISPR-Cas9 editing showed that perturbations in the autophagy pathway induce the accumulation of paxillin in the distal portion of the leading process of neuroblasts.

2) Confirm the impact of Atg5 on the density of neuroblasts in the R (increased?) and the OB (unchanged?).

We performed additional experiments to confirm the impact of autophagy deficiency on the density of neuroblasts in vivo. First, we used a new cohort of Atg5 WT and Atg5 cKO mice and induced recombination by a single tamoxifen injection. We obtained the same results and thus pooled the data. Our results showed that the density of neuroblasts in the RMS of Atg5 cKO mice increases and that this is associated with a concomitant significant decrease (p<0.05) in the density of neuroblasts in the olfactory bulb (OB) and RMS_OB_ (Figure 2M). Furthermore, we also analyzed the density of neuroblasts in the SVZ, RMS, RMS_OB_, and OB following the electroporation of Atg12 and LacZ (control) gRNAs. This analysis also showed that the density of Atg12-deficient neuroblasts increases in the RMS and decreases in the RMS_OB_ and OB (Figure 3L-M). These results suggest that a genetic perturbation of autophagy genes (Atg5, Atg12) in both the developing and adult RMS induces the accumulation of neuroblasts in the RMS proximal to the SVZ, which is accompanied by a significant decrease in the number of neuroblasts arriving in the RMS_OB_ and OB. These results provide further support for the involvement of autophagy-related genes in neuronal migration in vivo.

3) Confirm the loss of Atg12 protein.

We confirmed the loss of the Atg12 protein in the Atg12 gRNA experiments by performing immunolabeling for Atg12, as suggested by the reviewer. The images are shown in Figure 3G and the quantification data are given in the Results section.

4) Quantify autophagolysosomal activity (Figure 1I) by analyzing GFP+RFP/RFP punctae ratio.

To address this point, we performed new experiments using two-photon imaging of GFP+ and RFP+ punctae in migrating cells in acute brain sections. The two-photon imaging allowed us to obtain a better *z* resolution and thus more accurately quantify the dynamics of autophagosomes (GFP+/RFP+) with respect to autolysosomes (RFP+). These data are shown in Figure 1J-K.

In addition, while the science is strong, the novelty of the work is overstated. There is existing literature showing roles of autophagy in neuronal migration. The paper needs rewriting to accurately place your work in the context of prior research in the field.Specific recommendations include:i) Rewrite the Introduction and Discussion to cite the literature appropriately.ii) The second reviewer suggests that your Results could be presented in a different way in order to make better use of your data.

We have rewritten the Introduction and the Discussion and have also modified the flow of the Results section as suggested by reviewer #2. With regards to prior literature in terms of the role of autophagy in neuronal migration, we would like to point out that while all these studies show a link between neuronal migration and autophagy, the genetic manipulations used affected not only autophagy but also several other molecular pathways that may potentially impact cell migration (Gstrein et al., 2018; Peng et al., 2012; Petri et al., 2012; Li et al., 2018). For example, mutations in Vps15 and Vps18, in addition to the defects in autophagy, also perturb other vesicle transport pathways such as endocytosis (Peng et al., 2012; Gstrein et al., 2018), which also regulate neuronal migration (Cosker and Segal, 2014; Yap and Winckler, 2012). Let-7 and Foxp1 also affect several pathways (Petri et al., 2017; Li et al., 2018). As such, the exact roles played by key autophagy genes on the migration of neuronal cells are still elusive and requires further investigation. It is also unclear how autophagy is dynamically regulated in migrating neuroblasts to cope with the pace and periodicity of neuronal migration, whether its level and activity is modulated by migration-promoting and migration-inhibiting molecular cues, and how it is induced. We believe that our study addresses all these issues.